# Effect of *Epichloë gansuensis* Endophyte on Seed-Borne Microbes and Seed Metabolites in *Achnatherum inebrians*

Jinjin Liang,[a] Guoyu Gao,[a] Rui Zhong,[a] Bowen Liu,[a] Michael J. Christensen,[b] Yawen Ju,[c] Wu Zhang,[d] Yanzhong Li,[a] Chunjie Li,[a] Xingxu Zhang,[a] Zhibiao Nan[a]

aState Key Laboratory of Grassland Agro-Ecosystems, Center for Grassland Microbiome, Key Laboratory of Grassland Livestock Industry Innovation, Ministry of Agriculture and Rural Affairs, College of Pastoral Agriculture Science and Technology, Lanzhou University, Lanzhou, China
bGrasslands Research Centre, AgResearch, Palmerston North, New Zealand
cHuaiyin Institute of Agricultural Sciences of Xuhuai Region in Jiangsu, Jiangsu, China
dSchool of Geographical Science, Lingnan Normal University, Zhanjiang, China

Jinjin Liang and Guoyu Gao contributed equally to this work. Author order was determined in order of increasing seniority.

**ABSTRACT** The seed-borne microbiota and seed metabolites of the grass *Achnatherum inebrians*, either host to *Epichloë gansuensis* (endophyte infected [EI]) or endophyte free (EF), were investigated. This study determined the microbial communities both within the seed (endophytic) and on the seed surface (epiphytic) and of the protective glumes by using Illumina sequencing technology. *Epichloë gansuensis* decreased the richness of the seed-borne microbiota except for the epiphytic fungi of glumes and also decreased the diversity of seed-borne microbiota. In addition, metabolites of seeds and glumes were detected using liquid chromatography-mass spectrometry (LC-MS). Unlike with the seeds of EF plants, the presence of *E. gansuensis* resulted in significant changes in the content of 108 seed and 31 glume metabolites. A total of 319 significant correlations occurred between seed-borne microbiota and seed metabolites; these correlations comprised 163 (147 bacterial and 16 fungal) positive correlations and 156 (136 bacterial and 20 fungal) negative correlations. Meanwhile, there were 42 significant correlations between glume microbiota and metabolites; these correlations comprised 28 positive (10 bacterial and 18 fungal) and 14 negative (9 bacterial and 5 fungal) correlations. The presence of *E. gansuensis* endophyte altered the communities and diversities of seed-borne microbes and altered the composition and content of seed metabolites, and there were many close and complex relationships between microbes and metabolites.

**IMPORTANCE** The present study was to investigate seed-borne microbiota and seed metabolites in *Achnatherum inebrians* using high-throughput sequencing and LC-MS technology. *Epichloë gansuensis* decreased the richness of the seed-borne microbiota except for the epiphytic fungi of glumes and also decreased the diversity of seed-borne microbiota. Compared with endophyte-free plants, the content of 108 seed and 31 glume metabolites of endophyte-infected plants was significantly changed. There were 319 significant correlations between seed-borne microbiota and seed metabolites and 42 significant correlations between glume microbiota and metabolites.

**KEYWORDS** *Achnatherum inebrians*, *Epichloë gansuensis*, microbiome, metabolomics, seed-borne, metabonomics

*A*chnatherum inebrians is a perennial grass that is becoming increasing widespread in the highly modified arid/semiarid grasslands of northwest China (1–3). A feature of this invasive grass species is that nearly all plants are host to a seed-borne, mutualistic fungal endophyte, either *Epichloë gansuensis* (2, 4) or *Epichloë inebrians* (5). The presence of the *Epichloë* endophyte enhances tolerance to biotic stresses, such as insect pests (6) and pathogenic fungi (7, 8), and abiotic stresses, such as heavy metals (9), low temperature

Address correspondence to Wu Zhang, ldzw1987@163.com, or Xingxu Zhang, xxzhang@lzu.edu.cn.

The authors declare no conflict of interest.

(10), salt (11), drought (12), low N (13), and low P (14). One important advantage provided by the presence of the *Epichloë* endophyte is deterrence of grazing by livestock due to toxicity from the presence of alkaloids, leading to the increasing prevalence of this grass. This toxicity has led to the common name of drunken horse grass (DHG). Although *A. inebrians* is commonly regarded as a toxic weed, the presence of these tall plants provides protected space where increasingly rare endemic plants can reestablish and so help to overcome the degradation of these grasslands (3).

The focus of this study was to investigate the fungi and bacteria associated with the seeds of field-grown *A. inebrians* plants, both when host to an *Epichloë* endophyte (endophyte infected [EI]) and when endophyte free (EF). Previous studies of seed microbiomes have focused on potentially beneficial (15, 16) and pathogenic (17, 18) microorganisms. Bacteria and fungi on the surfaces of leaves, and other plant tissues, are referred to as epiphytic (19), and those inside leaves are referred to as endophytic (20). Seed-borne microbes are similar to those of leaves. Nelson summarized that the microbes of seeds can be located in internal seed tissues (endophytic species) and on the seed surface (epiphytic species) and that interactions among these microorganisms taking place in, on, and around seeds may have a subsequent impact on plant fitness and productivity (21). To gain additional understanding of the fungi and bacteria associated with EI and EF seeds of *A. inebrians*, the present study also examined the microbiota of the protective glumes that cover seeds.

In addition to gaining greater understanding of the fungi and bacteria associated with EI and EF seeds, the study examined the metabolites that they contained. In particular, were there differences in the metabolites present in the EI and EF seeds and also in the glumes from EI and EF seeds? Also, were there metabolites present that were known to be secondary metabolites of the *Epichloë* endophyte and of other fungi and bacteria? Another possibility was that within the seeds were chemicals produced by seed tissues in response to the presence of invading fungi and bacteria.

Plant metabolomics has a wide range of potential applications (22) and has become a key technical method to reveal plant genetic diversity (23) and to better understand plant development (24). It also has a role in investigating the adaptation mechanisms of plants to abiotic stresses, including drought (25, 26), salt stress (27, 28), and temperature (29), and to biotic stresses such as pathogenic fungi (30, 31). This study utilized liquid chromatography-mass spectrometry (LC-MS), as this technology (32) is particularly important for high-sensitivity and nontargeted plant metabolomics and is suitable for the rich diversity of metabolites in plants (33, 34). Secondary metabolites of plants play an important role in the interaction between endophytic fungi and host plants (35). Studies have shown that colonization by endophytic fungi affects the content of metabolites in host plants under different treatments (13, 14).

Previous studies involving DHG found that the presence of *E. gansuensis* influenced the root-associated fungal community under different growth conditions (36). Based on the morphological identification of spores, research has found that *E. gansuensis* alters the arbuscular mycorrhizal fungi (AMF) community of DHG rhizosphere soil under different growth conditions (37). Recently, Zhong et al. illustrated that *E. gansuensis* influences the AMF community of DHG roots and rhizosphere soil under different water conditions, as determined through the application of high-throughput sequencing technology (38). In addition, *E. gansuensis* and soil moisture also affect the bacterial diversity of roots and rhizosphere soil of DHG by altering rhizosphere soil properties (39). Meanwhile, plant growth-promoting rhizobacteria isolated from the rhizosphere soil of DHG can promote seed germination and seedling growth under salt stress (40). But few studies focused on how the presence of *E. gansuensis* affects the microbiome diversity and metabolites of DHG seeds.

In order to explore how the presence of the *E. gansuensis* endophyte affects the microbiome community and metabolites of DHG seeds, we collected EI and EF seeds from a field of the Yuzhong campus (Lanzhou University); these seeds were used to conduct high-throughput sequencing and LC-MS technology analysis to investigate the effects of *E. gansuensis* on microbiome diversity and metabolism of DHG seeds and glumes.

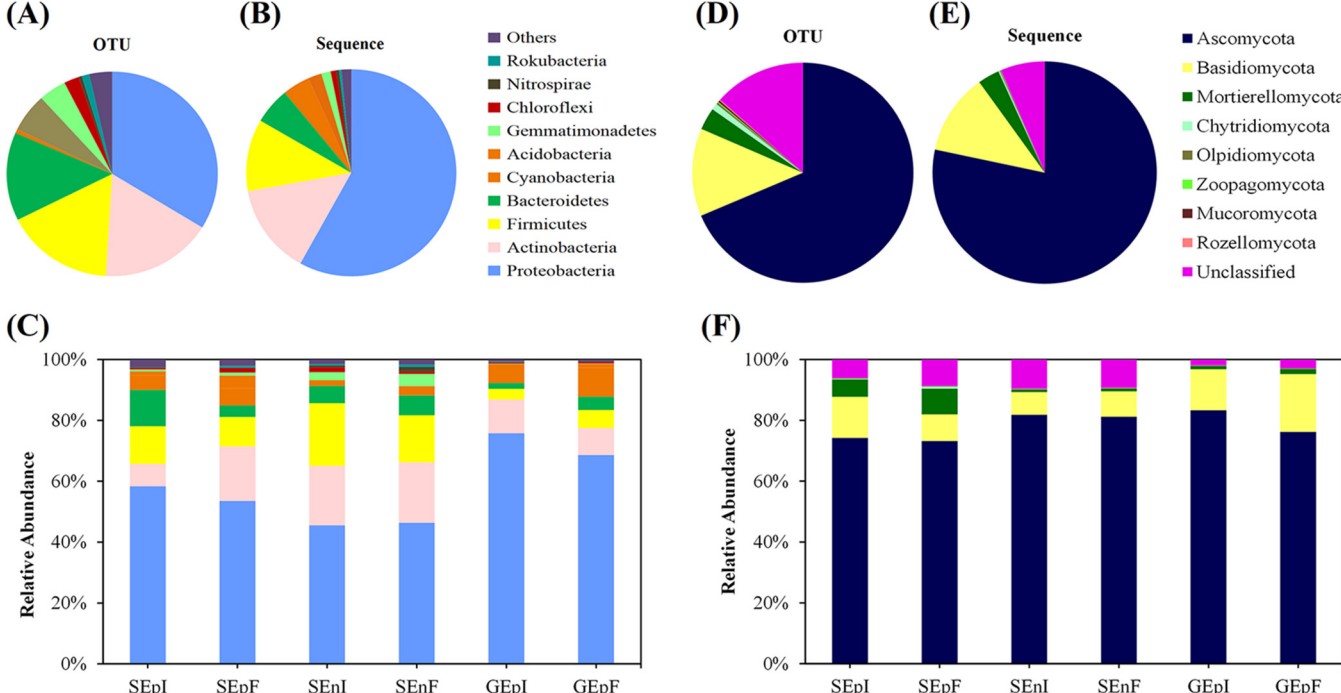

**FIG 1** Taxonomic composition of seed and glume microbial communities. OTUs of bacteria (A) and fungi (D) and sequences of bacteria (B) and fungi (E) in all samples and relative abundances of bacteria (C) and fungi (F) at the phylum level located on or in *Achnatherum inebrians* seeds and in the endophyte treatments SEpI (seed epiphytic microbes of the seed-borne of endophyte-infected plants), SEpF (seed epiphytic microbes of the seed-borne of endophyte-free plants), SEnI (seed endophytic microbes of the seed-borne of endophyte-infected plants), SEnF (seed endophytic microbes of the seed-borne of endophyte-free plants), GEpI (glume microbial community of endophyte-infected plants), and GEpF (glume microbial community of endophyte-free plants) (*n* = 8). See also Table S1 in the supplemental material.

We hypothesized that (i) *E. gansuensis* would alter the microbiome communities and diversities of *A. inebrians* seeds and glumes, (ii) *E. gansuensis* can influence the content and classes of metabolites in seeds and glumes, and (iii) there are complex and close correlations between seed-borne microbes and metabolites of EI and EF seeds and also between microbes and metabolites of EI and EF glumes.

## RESULTS

**Bacterial and fungal community compositions of seeds and glumes.** A total of 732,096 bacterial and 1,510,368 fungal sequences were obtained from seed and glume samples by high-throughput sequencing technology (see Table S1 in the supplemental material). All sequences were classified into 1,460 bacterial and 736 fungal operational taxonomy units (OTUs) at a 97% sequence similarity cutoff (Table S1). The 1,460 bacterial OTUs were divided into 11 phyla and 8 genera, and the 736 fungal OTUs were divided into 8 phyla and 10 genera. Among the bacterial communities of seed epiphytic microbes of endophyte-infected plants (SEpI), seed epiphytic microbes of endophyte-free plants (SEpF), seed endophytic microbes of endophyte-infected plants (SEnI), seed endophytic microbes of endophyte-free plants (SEnF), glume microbial community of endophyte-infected plants (GEpI), and glume microbial community of endophyte-free plants (GEpF), 498 OTUs were shared by all eight experimental plot sample repetitions (Fig. S2A; Table S2). In addition, in the fungal communities of SEpI, SEpF, SEnI, SEnF, GEpI, and GEpF, 377 OTUs were shared by all eight sample repetitions (Fig. S2B; Table S3). The bacterial and fungal rarefaction curves suggested that the 16S rRNA and internal transcribed spacer (ITS) gene sequences for all seed and glume samples reach the sequencing depths (Fig. S3A and B).

The relative abundances of bacteria and fungi at the phylum level of seed-borne communities were diverse among the three different parts of EI and EF seeds (Fig. 1; Table S1). The seed-borne bacterial communities were dominated by *Proteobacteria* (493 OTUs,

58% sequences), *Actinobacteria* (251 OTUs, 14% sequences), *Firmicutes* (243 OTUs, 11% sequences), and *Bacteroidetes* (204 OTUs, 5.7% sequences) (Fig. 1A and B; Table S1). The bacterial communities of seeds and glumes, at the genus level, were mostly dominated by *Pseudomonas* (15 OTUs, 3.6% sequences), *Bacillus* (10 OTUs, 2.2% sequences) *Sphingomonas* (9 OTUs, 4.9% sequences), and *Allorhizobium* (5 OTUs, 13.1% sequences) (Fig. S1A and B; Table S1). In addition, Ascomycota (505 OTUs, 78% sequences) was the most dominant phylum in the fungal community of seeds and glumes (Fig. 1D and E; Table S1); the next most dominant phyla were Basidiomycota (95 OTUs, 12% sequences) and Mortierellomycota (24 OTUs, 3% sequences) (Fig. 1D and E; Table S1). The most common genera of the fungal communities of seeds and glumes were *Penicillium* (27 OTUs, 2.0% sequences), *Mortierella* (18 OTUs, 2.8% sequences), and *Fusarium* (13 OTUs, 4.3% sequences) (Fig. S1D and E; Table S1).

**Bacterial and fungal community diversity of seeds and glumes.** The ACE index and the Chao index of seed-borne bacterial communities were significantly ($P < 0.05$) influenced by the presence of the *Epichloë* endophyte (Fig. 2A and C). Meanwhile, the three different parts of EI and EF seeds also had significant ($P < 0.05$) differences, as illustrated by the ACE index, Chao index, Shannon index, and Simpson index (Fig. 2A, C, E, and G, respectively). For the ACE index and Chao index, epiphytic and endophytic bacterial diversities of seeds were significantly higher than the bacterial diversity of the epiphytic bacteria of glumes (Fig. 2A and C). As the Shannon index illustrated, the diversity of endophytic bacterial communities of *A. inebrians* seeds was significantly higher than the diversity of the epiphytic bacteria of *A. inebrians* seeds and of the bacteria of glumes (Fig. 2E). Furthermore, the Simpson index of the epiphytic bacteria of glumes was higher than that of the epiphytic and endophytic bacteria of seeds (Fig. 2G).

The interaction between the *Epichloë* endophyte and the three different parts of EI and EF seeds had significant ($P < 0.05$) effects on the diversity of fungi of seeds and glumes, as summarized by the ACE index and Chao index (Fig. 2B and D). The treatments of different parts of EI and EF seeds had significant ($P < 0.05$) effects on seed fungal diversities, as summarized by the ACE index, Chao index, Shannon index, and Simpson index. The infection status of the *E. gansuensis* endophyte had a significant influence on the Simpson index.

Principal-coordinate analysis (PCoA) showed that the bacterial community compositions of seeds and glumes between EI and EF DHG significantly ($P < 0.05$) differed among the different parts of EI and EF seed treatments (Fig. 3A; Table 1). In addition, *E. gansuensis* endophyte infection status, different parts of EI and EF seeds, and their interactions had significant ($P < 0.05$) effects on the diversity of the fungal communities of seeds and glumes of *A. inebrians* (Fig. 3B; Table 1).

**Effect of *Epichloë* endophytes on metabolites of seeds and glumes.** In total, 517 and 517 metabolites were successfully detected from EI and EF seeds and glumes of *A. inebrians*, respectively (Tables S4 and S5). This study calculated the orthogonal projections to latent structures discriminant analysis (OPLS-DA) and principal-component analysis (PCA) on these metabolites of seeds (Fig. S4) and glumes (Fig. S5). The OPLS-DA models of seeds (R2X = 0.609, R2Y = 1, Q2Y = 0.976) and of glumes (R2X = 0.597, R2Y = 0.94, Q2Y = 0.804) are shown in Fig. S4A and S5A, respectively. The PCA results showed that these detected metabolites of seeds (PC1 = 40.67%, PC2 = 28.50%) (Fig. S4C) and glumes (PC1 = 88.52%, PC2 = 7.18%) (Fig. S5C) were significantly diverse between EI and EF seeds and glumes. All showed that there were differences in detected metabolites between EI and EF seeds and glumes (Fig. S4A and S5A).

Based on the fold change (FC) of >2, variable importance in the project (VIP) of >1, and *P* of 0.05, 108 and 31 different metabolites between EI and EF seeds and glumes, respectively, were separated (Tables S6 and S7). Among metabolites of EI seeds, 69 were upregulated and 39 were downregulated (Fig. S4B; Table S6); among metabolites of EI glumes, 25 were upregulated and 6 were downregulated (Fig. S5B; Table S7).

In seeds, the 108 differential metabolites could be divided into 10 classes; the 7 major classes included alkaloids, lipids, others, phenolic acids, organic acids, amino acids and derivatives, and nucleotides and derivatives (Table 2). In glumes, these 31 differential

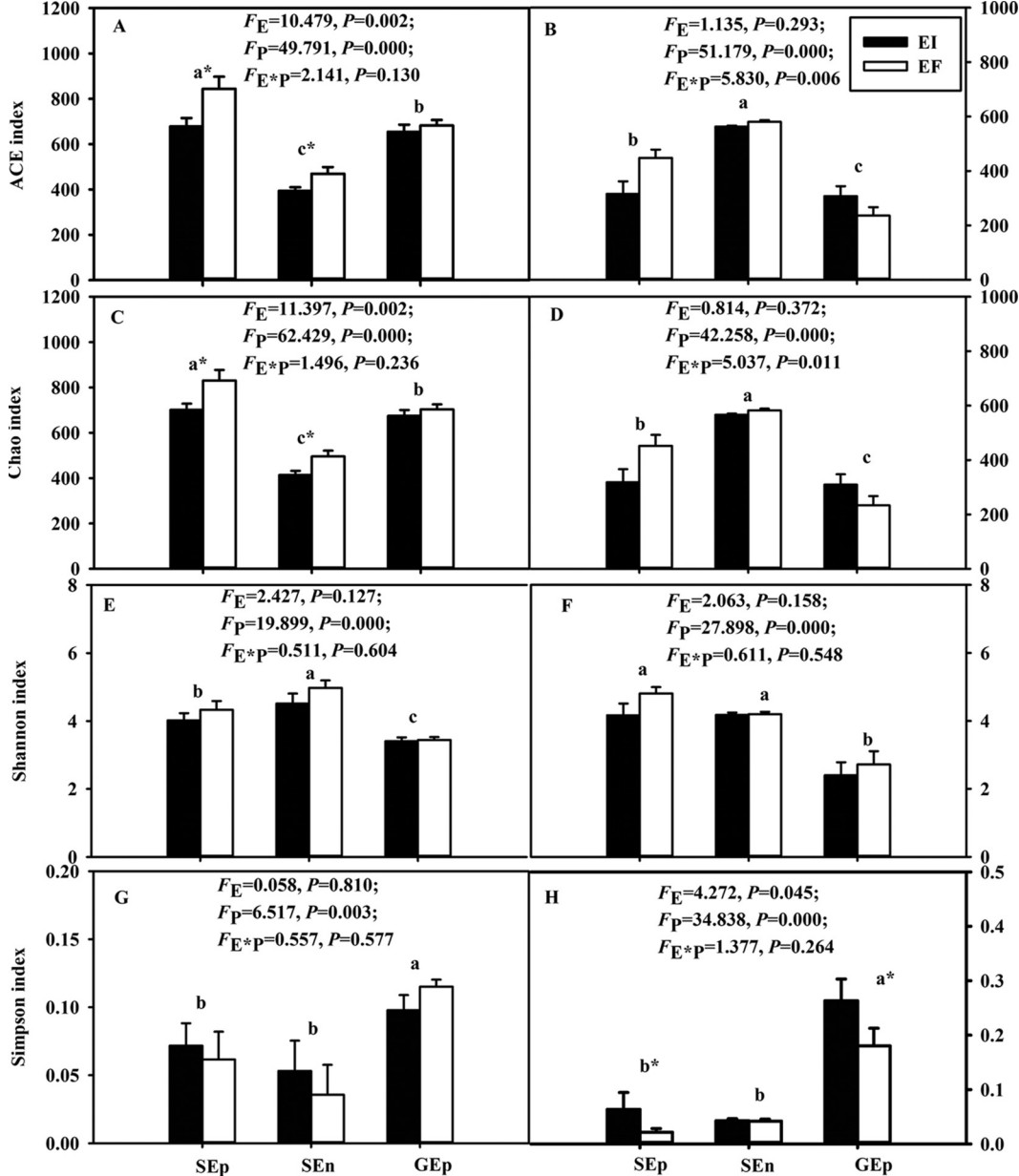

**FIG 2** Seed and glume microbial community alpha diversity index. Bacterial (A, C, E, and G) and fungal (B, D, F, and H) alpha diversities in and on seeds and in glumes with and without endophyte. E, *Epichloë gansuensis* endophyte infection status; P, different parts of EI and EF seeds; EI, endophyte infected; EF, endophyte free; Sep, seed epiphytic; SEn, seed endophytic; GEp, glume epiphytic (*n* = 8). Values are means ± standard errors of the mean (SEMs), with bars indicating SEs. An asterisk after a lowercase letter indicates significant difference at a *P* of <0.05 (independent *t* test) between EI and EF plants at seed and glume at 0.05 level. Different lowercase letters mean a significant difference at a *P* of <0.05 between seeds and glumes at 0.05 level.

metabolites could be divided into 6 classes; the 3 major classes were alkaloids, others, and organic acids (Table 2).

Compared with EF seeds, a total of 69 upregulated and 39 downregulated differential metabolites were detected in EI seeds (Table S6); the top 10 up- and downregulated metabolites are shown in Table 3. Compared with EF glumes, a total of 25 upregulated and 6 downregulated differential metabolites were detected in EI glumes (Table S7); the top 5 differentially accumulated metabolites are shown in Table 4. In seeds, diethylcarbamazine, 3,4-dimethoxycinnamic acid, methylergonovine, 3-isobutyl-1-methylxanthine (IBMX), hexacosanoic acid, nortriptyline, α-asarone, 6-benzylaminopurine, (−)-tylocrebrine, and D-biotin were significantly (*P* < 0.05) increased compounds, while the compounds that decreased

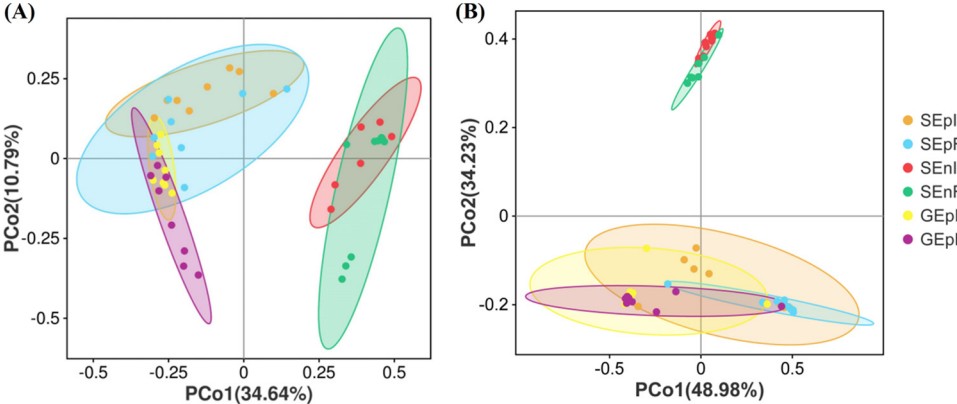

**FIG 3** Bacteria (A) and fungi (B) of seeds and glumes. PCoA was based on Bray-Curtis dissimilarities at the OTU level ($n = 8$).

significantly ($P < 0.05$) were nicotinamide, linoleoyl ethanolamide, 3-hydroxy-4-methoxycinnamic acid, DL-benzylsuccinic acid, lithocholic acid, eicosapentaenoic acid ethyl ester, alpha-linolenoyl ethanolamide, (+)-alpha-pinene, stearidonic acid, and minoxidil (Table 3). In glumes, hexacosanoic acid, diethylcarbamazine, vitamin $D_2$ (ergocalciferol), dihydrotachysterol, and perillyl alcohol were significantly ($P < 0.05$) increased compounds, while the compounds that decreased significantly ($P < 0.05$) were 9-decen-1-ol, 25-hydroxyvitamin $D_3$, 4,6-dioxoheptanoic acid, 2-methylglutaric acid, and pyrocatechol (Table 4).

The study analyzed different metabolites according to the Kyoto Encyclopedia of Genes and Genomes (KEGG) database; the results showed that a total of 34 and 9 metabolites were annotated by KEGG in seeds and glumes, respectively (Tables S6 and S7). There were 20 metabolite pathways significantly enriched ($P < 0.05$) in EI seeds (Fig. S6A), and 11 metabolites pathways were significantly enriched ($P < 0.05$) in EI glumes (Fig. S6B). In seeds, KEGG analysis showed that fatty acid biosynthesis and sesquiterpenoid and triterpenoid biosynthesis were major enriched biological pathways (Fig. S6A). In glumes, KEGG analysis showed that the most enriched biological pathways were monoterpenoid biosynthesis, limonene and pinene degradation, D-arginine and D-ornithine metabolism, steroid biosynthesis, phenylalanine tyrosine and tryptophan biosynthesis, nicotinate and nicotinamide metabolism, pyrimidine metabolism, and tryptophan metabolism (Fig. S6B).

**Correlation network between metabolites and microbes in seeds and glumes.** To explore correlations between metabolites and microorganisms, we analyzed the

**TABLE 1** ANOSIM and PERMANOVA analysis of differences in bacterial and fungal community compositions in and on seeds and on glumes of EI and EF *A. inebrians* as calculated by amplicon sequencing[a]

| Community and treatment | df[b] | PERMANOVA (Bray-Curtis) | | ANOSIM (Bray-Curtis) | |
|---|---|---|---|---|---|
| | | F | P | R | P |
| Bacteria | | | | | |
| E | 1 | 3.2724 | **0.014** | 0.1835 | **0.001** |
| P | 2 | 14.687 | **0.000** | 0.6539 | **0.000** |
| E × P | 2 | 1.5393 | | 0.1396 | |
| | | | | | |
| Fungi | | | | | |
| E | 1 | 2.889 | 0.073 | 0.3160 | **0.000** |
| P | 2 | 35.103 | **0.000** | 0.6627 | **0.000** |
| E × P | 2 | 2.961 | **0.042** | | |

[a]ANOSIM, statistical test of similarity; PERMANOVA, permutational mutlivariate two-way analysis of variance. E, *Epichloë gansuensis* endophyte infection status; P, different parts of EI and EF seeds. F value is used to measure the influence of a variable on the dependent variable in multivariate analysis of variance. R value represents the degree of linear correlation between the two variables. Boldface values indicate $P < 0.05$.
[b]df, degrees of freedom.

**TABLE 2** Classification of 108 and 31 detected differential metabolites in *A. inebrians* seeds and glumes, respectively

| Class | No. of compounds in: | |
|---|---|---|
| | Seeds | Glumes |
| Alkaloids | 21 | 9 |
| Lipids | 21 | 3 |
| Others | 16 | 7 |
| Phenolic acids | 15 | |
| Organic acids | 11 | 7 |
| Amino acids and derivatives | 10 | 3 |
| Nucleotides and derivatives | 10 | 2 |
| Terpenoids | 2 | |
| Flavonoids | 1 | |
| Lignans and coumarins | 1 | |
| Total | 108 | 31 |

Spearman's correlation networks between metabolites that were significantly up- or downregulated in seeds and seed-borne microbial phyla (Fig. 4; Tables S8 to S13).

There were 135 and 3 significant ($P < 0.05$) correlations between seed differential metabolites and seed epiphytic bacterial (Fig. 4A; Table S8) and fungal (Fig. 4B; Table S9) phyla, respectively. The main microbial phyla involved with these correlations included *Acidobacteria*, *Actinobacteria*, *Bacteroidetes*, *Chloroflexi*, *Proteobacteria*, Basidiomycota, and Mucoromycota. In comparison with SEpF, the relative abundances of *Bacteroidetes*, *Proteobacteria*, and Basidiomycota in SEpI were increased but those of other phyla were decreased (Table S1). In the overall network diagram, the bacteria in the *Acidobacteria* and *Actinobacteria* were most closely related to the metabolites, with the degrees (degree represents the number of phyla or metabolites connected to other metabolites or phyla) of *Acidobacteria* and *Actinobacteria* being 48 and 35, respectively. The phyla next most closely related to metabolites were *Bacteroidetes*, *Chloroflexi*, and *Proteobacteria*, with 16, 14, and 10 correlations with metabolites, respectively, and these correlations between the above phyla and metabolites were all positive (Fig. 4A; Table S8). In addition, the fungi in the Basidiomycota were negatively correlated with Tyr-Met and myristic acid (Fig. 4B; Table S9). In whole bacterial and fungal networks, there were two key metabolites; Tyr-Met was positively correlated with *Actinobacteria* and *Acidobacteria* but negatively

**TABLE 3** Top 10 differentially accumulated metabolites in EI seeds compared to EF seeds of *A. inebrians*[a]

| Compound | Class | No. of SEI | No. of SEF | FC | Log$_2$ FC | Up- or downregulation |
|---|---|---|---|---|---|---|
| Diethylcarbamazine | Alkaloids | $9.59 \times 10^{-4}$ | $1.76 \times 10^{-2}$ | 18.32 | 4.49 | Up |
| 3,4-Dimethoxycinnamic acid | Lipids | $2.46 \times 10^{-4}$ | $1.49 \times 10^{-2}$ | 60.69 | 6.41 | Up |
| Methylergonovine | Alkaloids | $2.06 \times 10^{-4}$ | $1.49 \times 10^{-2}$ | 72.03 | 6.63 | Up |
| IBMX | Nucleotides and derivatives | $1.54 \times 10^{-4}$ | $7.56 \times 10^{-3}$ | 48.53 | 6.06 | Up |
| Hexacosanoic acid | Organic acids | $2.03 \times 10^{-4}$ | $6.80 \times 10^{-3}$ | 3.34 | 1.73 | Up |
| Nortryptyline | Lipids | $9.16 \times 10^{-5}$ | $4.80 \times 10^{-3}$ | 52.31 | 5.86 | Up |
| $\alpha$-Asarone | Phenolic acids | $8.28 \times 10^{-4}$ | $4.20 \times 10^{-3}$ | 5.07 | 2.54 | Up |
| 6-Benzylaminopurine | Nucleotides and derivatives | $1.88 \times 10^{-4}$ | $3.50 \times 10^{-3}$ | 18.61 | 4.62 | Up |
| (−)-Tylocrebrine | Alkaloids | $1.75 \times 10^{-4}$ | $3.06 \times 10^{-3}$ | 17.27 | 4.09 | Up |
| D-Biotin | Others | $1.15 \times 10^{-4}$ | $2.9 \times 10^{-3}$ | 24.82 | 5.73 | Up |
| Nicotinamide | Others | $1.04 \times 10^{-2}$ | $1.72 \times 10^{-3}$ | 0.17 | −2.62 | Down |
| Linoleoyl ethanolamide | Lipids | $2.98 \times 10^{-3}$ | $1.45 \times 10^{-3}$ | 0.49 | −1.04 | Down |
| 3-Hydroxy-4-methoxycinnamic acid | Phenolic acids | $2.51 \times 10^{-3}$ | $1.24 \times 10^{-3}$ | 0.49 | −0.99 | Down |
| DL-Benzylsuccinic acid | Organic acids | $3.393 \times 10^{-3}$ | $1.18 \times 10^{-3}$ | 0.35 | −1.52 | Down |
| Lithocholic acid | Organic acids | $2.26 \times 10^{-3}$ | $9.45 \times 10^{-4}$ | 0.42 | −1.25 | Down |
| Eicosapentaenoic acid ethyl ester | Lipids | $8.92 \times 10^{-4}$ | $4.13 \times 10^{-4}$ | 0.46 | −1.10 | Down |
| Alpha-linolenoyl ethanolamide | Alkaloids | $8.83 \times 10^{-4}$ | $3.90 \times 10^{-4}$ | 0.44 | −1.18 | Down |
| (+)-Alpha-pinene | Terpenoids | $8.26 \times 10^{-4}$ | $2.51 \times 10^{-4}$ | 0.30 | −1.75 | Down |
| Stearidonic acid | Lipids | $6.15 \times 10^{-4}$ | $2.49 \times 10^{-4}$ | 0.40 | −1.31 | Down |
| Minoxidil | Nucleotides and derivatives | $1.57 \times 10^{-3}$ | $2.47 \times 10^{-4}$ | 0.16 | −2.23 | Down |

[a]SEI, seed metabolites of endophyte-infected plants; SE, seed metabolites of endophyte-free plants ($n = 6$).

**TABLE 4** Top five differentially accumulated metabolites in EI glumes compared to EF glumes of *A. inebrians*[a]

| Compound | Class | GEI | GEF | FC | Log$_2$ FC | Up- or downregulation |
|---|---|---|---|---|---|---|
| Hexacosanoic acid | Organic acids | $3.72 \times 10^{-3}$ | $1.59 \times 10^{-2}$ | 4.27 | 2.05 | Up |
| Diethylcarbamazine | Alkaloids | $7.45 \times 10^{-4}$ | $7.55 \times 10^{-3}$ | 10.13 | 3.73 | Up |
| Vitamin D$_2$ (ergocalciferol) | Others | $1.89 \times 10^{-3}$ | $6.73 \times 10^{-3}$ | 3.56 | 1.80 | Up |
| Dihydrotachysterol | Others | $6.41 \times 10^{-4}$ | $5.40 \times 10^{-3}$ | 8.43 | 3.01 | Up |
| Perillyl alcohol | Others | $1.24 \times 10^{-3}$ | $4.55 \times 10^{-3}$ | 3.66 | 1.83 | Up |
| 9-Decen-1-ol | Others | $9.86 \times 10^{-3}$ | $3.99 \times 10^{-3}$ | 0.41 | −1.23 | Down |
| 25-Hydroxyvitamin D$_3$ | Others | $2.17 \times 10^{-3}$ | $1.00 \times 10^{-4}$ | 0.46 | −1.16 | Down |
| 4,6-Dioxoheptanoic acid | Organic acids | $7.01 \times 10^{-4}$ | $3.32 \times 10^{-4}$ | 0.47 | −1.05 | Down |
| 2-Methylglutaric acid | Organic acids | $8.38 \times 10^{-4}$ | $3.28 \times 10^{-4}$ | 0.39 | −1.37 | Down |
| Pyrocatechol | Others | $5.29 \times 10^{-4}$ | $2.43 \times 10^{-4}$ | 0.46 | −1.14 | Down |

[a]GEI, glume metabolites of EI plants; GEF, glume metabolites of EF plants ($n = 6$).

correlated with Basidiomycota, and myristic acid was positively correlated with *Rokubacteria* but negatively correlated with Basdiomycota (Fig. 4; Tables S8 and S9).

There were 148 and 33 significant ($P < 0.05$) correlations between differentially accumulated metabolites of seeds and seed endophytic bacterial (Fig. 4C; Table S10) and fungal (Fig. 4D; Table S11) phyla, respectively. The main microbial phyla involved with these correlations included *Chloroflexi*, *Firmicutes*, *Nitrospirae*, and Basidiomycota. In comparison with SEnF, the relative abundances of *Chloroflexi* and *Firmicutes* in SEnI were increased but other phyla were decreased (Table S1). In the network diagram, the bacteria in the *Chloroflexi*, *Firmicutes*, and *Nitrospirae* were mostly correlated with metabolites, with the highest degree being 38; the phylum next most closely related to metabolites was *Cyanobacteria*, which had 24 correlations with metabolites (Fig. 4C; Table S10). Furthermore, fungi in the Basidiomycota were most closely related to 22 metabolites (Fig. 4D; Table S11). In whole bacterial and fungal networks, 6-hydroxymelatonin was a key metabolite, positively correlated with *Firmicutes* and negatively correlated with *Gemmatimonadetes*, *Chloroflexi*, and *Rokubacteria* (Fig. 4; Tables S10 and S11).

There were 19 and 23 significant ($P < 0.05$) correlations between glume differential metabolites and glume bacterial (Fig. 4E; Table S12) and fungal (Fig. 4F; Table S13) phyla, respectively. The main microbial phyla involved with these correlations included *Bacteroidetes* and Ascomycota. In comparison with GEpF, the relative abundance of Ascomycota was increased and *Bacteroidetes* was decreased in GEpI (Table S1). In the network diagram, bacteria in the *Bacteroidetes* were mostly correlated with 8 metabolites (Fig. 4E; Table S12). Moreover, the fungi in the Ascomycota were closely related with 23 glume metabolites, and 18 out of the 23 correlations were positive (Fig. 4F; Table S13).

## DISCUSSION

The present study investigated the influence of the *E. gansuensis* endophyte on seed-borne microbial communities and seed metabolites of DHG. In particular, this study performed correlation analysis between seed-borne microbes and seed metabolites and found that there were many close and complex correlations between them.

**Effect of *E. gansuensis* on seed-borne microbial communities.** Results revealed that seed-borne bacterial communities were mostly dominated by phyla such as *Proteobacteria*, *Actinobacteria*, *Firmicutes*, and *Bacteroidetes*, and the core genera were *Pseudomonas*, *Bacillus*, *Sphingomonas*, and *Allorhizobium*. These results were similar to those of the study by Bastías et al., which looked at *Lolium multiflorum* seeds infected by *Epichloë occultans* (41). Previous studies reported that seed-borne fungal communities were dominated by Ascomycota and Basidiomycota (42–44), and those results were in line with the present study. Furthermore, *Penicillium*, *Mortierella*, and *Fusarium* were prominent genera of seed-borne fungal communities of DHG.

Bastías et al. not only demonstrated that *Epichloë* endophytes affected the composition and diversity of the seed microbiota, but they also considered that the bacterial microbes associated with leaf tissues of mother plants are the main source of bacteria for seeds (41). There were many studies that proved that *Epichloë* endophytes changed

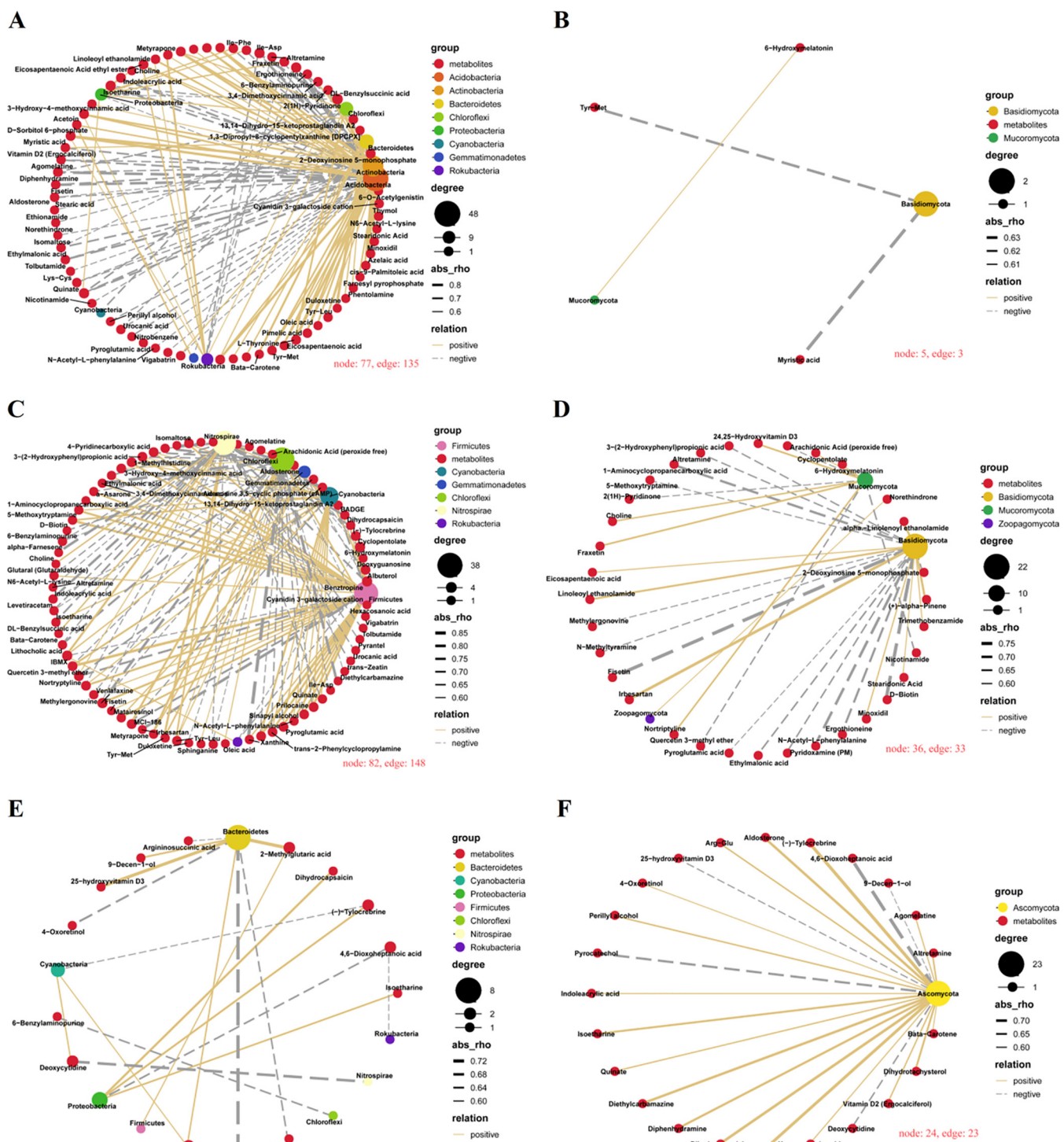

**FIG 4** Spearman's correlation networks between seed differential metabolites and seed-borne microbial phyla. (A) Seed differential metabolites and seed epiphytic bacterial phyla; (B) seed differential metabolites and seed epiphytic fungal phyla; (C) seed differential metabolites and seed endophytic bacterial phyla; (D) seed differential metabolites and seed endophytic fungal phyla; (E) glume differential metabolites and glume epiphytic bacterial phyla; (F) glume differential metabolites and glume epiphytic fungal phyla. node, metabolites or phyla; edge, correlations between metabolites and phyla; group, classification of metabolites and phyla; degree, edge number of one node; abs, extent of correlation; relation, yellow line indicates positive correlation between metabolites and phyla and gray dotted line indicates negative correlation.

the foliar microbiome community of host plants; for example, the presence of *Epichloë coenophiala* in tall fescue (*Festuca arundinacea*) modified the leaf-associated fungal communities and the presence of *Epichloë festucae* var. *lolii* modified the bacterial communities associated with seedlings of perennial ryegrass (*Lolium perenne*) plants

(45, 46). A recent study found that the presence of *Epichloë* changed the phyllosphere microbial communities on *A. inebrians* (47). In accordance with previous studies, the current study's findings provide strong support for the presence of an *Epichloë* endophyte modifying the bacterial microbes associated with the foliar tissues and seeds of mother plants.

The presence of *E. gansuensis* changed the composition and diversity of seed-borne microbes of DHG, which proved the first hypothesis, i.e., that *E. gansuensis* would modulate the microbial diversity of DHG seeds and glumes. Previous studies also showed that *Epichloë* endophytes had impacts on the microbial diversity of host plants; for example, the presence of *E. occultans* in *Lolium multiflorum* seeds resulted in a higher diversity in the bacterial community than did the presence of endophyte-free seeds (41). Similarly, *E. gansusensis* increased root-associated AMF diversity under drought conditions of DHG (36). *E. gansuensis* decreased the Shannon diversity of the root-associated bacterial community but increased the Shannon diversity of the rhizosphere soil bacterial community of *A. inebrians* (39). Recently, a study found that the *Epichloë* endophyte increased the diversity of phyllosphere bacterial and fungal communities on *A. inebrians* leaves (47). Results from the present study showed that *E. gansuensis* decreased bacterial and fungal diversity and richness except for the GEpl richness. *Epichloë* endophytes may modulate the fungal and bacterial diversity of seed-borne microbial communities by modifying certain plant physiological responses.

**Seed-borne epiphytic, endophytic, and glume microbial community diversity.** This study showed that the decreasing order of the alpha diversity index in DHG seed-borne bacteria was from epiphytic, to endophytic, to glume epiphytic and that the decreasing order of the alpha diversity index in DHG seed-borne fungi was from endophytic, to epiphytic, to glume epiphytic. A previous study showed that microorganisms were more abundant in the seed coat than in the endosperm and the embryo (48). Nelson summarized that it is essential to distinguish between epiphytic and endophytic microbes when work is carried out on the microbiomes of seeds (21). The endophytic microbes of seeds have typically been considered to be composed of commensals or mutualists (16, 43, 49, 50), and the epiphytic microbe communities were dominated by species of well-known plant pathogens of the genera *Fusarium*, *Phoma*, *Pyrenophora*, *Alternaria*, and *Leptosphaeria* (21, 51). In studies of the cultivable endophytic bacteria, abundance estimates may range from $10^1$ to $10^2$ CFU/g seed (52, 53) to as high as $10^6$ to $10^8$ CFU/g seed (51, 54). Wang found that the diversity of epiphytic bacteria was higher than that of endophytic bacteria in tomato seeds (55). Most seed-borne fungi were located in the endosperm, with decreasing numbers in the pericarp and embryo in *L. perenne* seeds, as determined by using a component plating method (56). For the sites of the fungi isolated from *Sorghum sudanense* seeds, the order of decreasing frequency of fungal species was seed coat, pericarp, endosperm, and embryo (57).

Recently, Bastías et al. reported that the presence of *Epichloë* endophytes increased the population of *Pseudomonas*, and these results were in line with the present study (41). Bacteria of the genus *Pseudomonas* have bene reported to enhance plant growth via production of auxins, phosphate-solubilizing compounds, and other growth-promoting compounds (58). *Pseudomonas stutzeri* increased *Oryza sativa* seedling resistance under salt stress via production of exopolysaccharide (59). A *Pseudomonas* strain protected *O. sativa* and *Poa annua* seedlings from fungal pathogens via the production of antimicrobials (60). The present study also showed that *E. gansuensis* increased the relative abundance of *Sphingomonas*. A strain of *Sphingomonas melonis* accumulated and was transmitted across generations in disease-resistant rice seeds, conferring resistance to disease-susceptible phenotypes by producing anthranilic acid (16). The present study also found that the presence of *E. gansuensis* decreased the abundance of fungi, including *Mortierella* and *Fusarium* species. Of importance is that many *Fusarium* species are pathogens of a wide range of plants (21, 61). Based on the findings of this and other studies, the presence of *E. gansuensis* seems likely to enhance the resistance of DHG plants to some fungal diseases via increases in the amount of beneficial bacteria and decreases in the presence of pathogenic fungi associated with seeds.

**Effect of the *E. gansuensis* endophyte on seed metabolites.** The present study showed that infection of DHG by *E. gansuensis* changed the class and content of metabolites of seeds and glumes, which completely supported our second hypothesis. Importantly, metabolomics plays a vital role in tracking temporal changes in metabolites through the entire growth cycle of plants (62–64) and through exposure to stress (27, 28). In many studies, researchers conducted experiments to address the effects of *Epichloë* endophytes on metabolites of host plants, and these mainly focused on alkaloids (65), acids (organic acids, amino acids, and fatty acids) (13), and amino acids (14).

The absence of alkaloids in EF plants is the biggest difference between EI and EF DHG. Alkaloids, including indol-diterpene, pyrrolapyrazine, pyrrolizidine, and ergot alkaloids, can be detected in *Epichloë* endophyte-infected plants (66–68). In 1984, the first finding was made of the presence of ergonovine and ergonovinine in DHG plants (69). In 1996, it was reported that the presence of these two alkaloids in DHG plants was associated with the presence of a seed-borne *Epichloë* endophyte (70). The alkaloid content, and in particular that of ergine and ergonovine, was linked with the growth of DHG plants, and frequent cutting (fortnightly) at a mowing height of 7.5 cm resulted in a significant boost to the content of the bioactive, endophyte-produced alkaloids (71). Further, studies found that the content of alkaloids increased over the DHG growing period under the abiotic stresses of salt and drought, and the content of cytotoxic ergonovine in particular increased under these stress conditions (72, 73). A recent study found that metabolites other than ergonovine and ergine, including epichlicin and cyclosporine T, isolated from host plants with an *Epichloë* endophyte, had significant antifungal and anti-insect activity (74).

Metabolomics results found that the presence of *E. gansuensis* influenced a wide range of metabolites produced in seeds or glumes, including purine derivatives, tryptamine derivatives, lignins, and aldehyde in DHG seeds and sterols and phenols in DHG glumes. Other studies on host grasses with an *Epichloë* endophyte also focused on metabolites in general, rather than just alkaloids. A study found that *Neotyphodium uncinatum* (=*Epichloë uncinata*) endophytes in host plants may affect soil insect distribution by altering the presence of root volatiles that affect insect behavior (75). *Epichloë coenophiala* in tall fescue can affect root exudate composition, including lipids, carbohydrates. and carboxylic acids, affecting plant growth (76). Furthermore, Hou et al. reported that *E. gansuensis* increased the tolerance of DHG to low N stress by increasing the content of glucose-6-phosphate and organic acids in leaves and increasing the content of fatty acids and amino acids in roots (13). Meanwhile, the presence of *Epichloë* endophytes improves DHG growth by regulating the metabolic pathway of amino acids, amino acid content, and organic acid content at low P stress (14). Metabolites from alkaloids to nonalkaloids produced by DHG seeds and glumes of *E. gansuensis*-hosting DHG plants are equally important because the defensive mutualism between grasses and the *Epichloë* endophytes involves a wider range of metabolites than just the recognized endophyte-produced alkaloids.

**Impact factors on the microbes of seeds and glumes.** Soil possesses high microbial diversity, and this directly and indirectly influences seed microbes (75, 76). A recent study showed that many soil bacteria could reach the leaves and flowers of *Arabidopsis thaliana* (77). Flowers are the origin of seed development and dispersal and also contain a large microbiome (78), and Nelson showed that microbes associated with flowers are an important source of seed-associated microbes (21). Furthermore, the floral nectar, which comprises secondary metabolites, particularly sugars and amino acids, can provide a source of energy and available carbon for seed-borne microbial growth (79).

The present study found that there were complex and close correlations between seed-borne microbes and seed metabolites or correlations between glume microbes and metabolites, which partially supported that there are complex and close correlations between seed-borne microbes and metabolites between EI and EF seeds and also between glume microbes and metabolites between EI and EF glumes. The presence of *E. gansuensis* changes the seed-borne microbiota and likely increases and decreases the number of beneficial bacteria and pathogenic fungi, respectively. These changes resulting from the presence of *E. gansuensis* result in the enhanced presence

of beneficial antibiotic products, auxin, and alkaloids that promote the growth and persistence of host plants. This study has confirmed that there are close and complex relationships between seed-borne microbes and seed metabolites and so enhances the understanding of how seed-associated microbes interact with each other and play an important role in the growth and development of the following generations.

## MATERIALS AND METHODS

**Site description and experimental design.** The EI and EF seeds of *A. inebrians* were collected from an experimental field in Yuzhong campus (104°39′E, 35°89′N; altitude, 1,653 m) of the College of Pastoral Agriculture Science and Technology of Lanzhou University. These EI and EF plots, also used in other studies, were established in 2017 and managed as described by Liu et al. (47). From 2018 to 2019, the leaf sheaths and seeds of all plants were stained with aniline blue (2) to check the endophyte infection status and to ensure that the selected seeds for this study were 100% and 0% infected, respectively.

**Sampling.** Eight experimental plots were randomly selected in September 2019 for the collection of seeds for the present study. For each subplot, three mature plants were selected for the collection of seeds, and then these seed samples were mixed into a composite sample. The collected seeds were stored at 4℃ in a refrigerator, from which we later randomly selected 50 EI and EF seeds. We wore sterile gloves and separated glumes from seeds on an ultraclean worktable sterilized with UV light for 1 h to ensure the presence of sterility throughout the separation process. The glumes and seeds removed from the seeds were immediately stored in sterile tubes, frozen in liquid nitrogen for 3 h, and then stored in a −80℃ freezer for subsequent sequencing.

A total of 16 seed samples (eight EI samples, eight EF samples) were used for the detection of seed epiphytic and endophytic microbes. There were 16 glume samples (eight EI samples, eight EF samples) used for the detection of glume epiphytic microbes.

Seed epiphytic microbes were washed from seed surfaces. Fifty seeds were transferred into 50-mL plastic tubes filled with 30 to 40 mL phosphate-buffered saline (PBS) buffer, along with two blank controls without added seeds, followed by oscillation for 30 to 60 min at 150 to 200 rpm, sonication for 5 min, and further oscillation for 30 to 60 min at 150 to 200 rpm. The seeds were removed, and the suspension was centrifuged at 10,000 $\times$ $g$ for 10 min to obtain precipitates containing bacteria, fungal spores, and hyphae dislodged from the surface of the seeds.

For seed endophytic microbes, the above-described harvested seeds were surface sterilized in 1% NaClO (sodium hypochlorite) for 3 min, followed by 70% ethanol for 10 min, and rinsed with sterilized water five times.

Glume epiphytic microbes were washed from the glume surfaces. Fifty glumes were transferred into 50-mL plastic tubes filled with 30 to 40 mL PBS buffer, along with two blank controls without added glumes, followed by oscillation for 30 to 60 min at 150 to 200 rpm, sonication for 5 min, and further oscillation for 30 to 60 min at 150 to 200 rpm. The glumes were removed, and the suspension was centrifuged at 10,000 $\times$ $g$ for 10 min to obtain precipitates containing bacteria, fungal spores, and hyphae dislodged from the surface of the glumes. Subsequently, these separated seed and glume samples were stored at −80℃ in a freezer before DNA extraction and metabolite detection.

**DNA extraction, PCR amplification, and sequencing.** DNA of these seed and glume samples was extracted using an MN NucleoSpin 96 soil DNA kit according to the manufacturer's instructions. The 16S rRNA genes were amplified using the universal primers 335F (5′-CADACTCCTACGGGAGGC-3′) and 769R (5′-ATCCTGTTTGMTMCCCVCRC-3′). The internal transcribed spacer (ITS1) genes were amplified using the universal primers ITS1F (5′-CTTGGTCATTTAGAGGAAGTAA-3′) and ITS2 (5′-GCTGCGTTCTTCATCGATGC-3′). PCR amplification was performed in a total volume of 50 $\mu$L, which contained 10 $\mu$L buffer, 0.2 $\mu$L Q5 high-fidelity DNA polymerase, 10 $\mu$L high GC enhancer, 1 $\mu$L deoxynucleoside triphosphate, 10 $\mu$M each primer, and 60 ng genomic DNA. The PCRs were performed using the following cycling conditions: 98℃ for 30 s, followed by 10 cycles of 98℃ for 10 s, 65℃ for 30 s, and 72℃ for 30 s, and a final step of 72℃ for 5 min. The reactions were run on 1.8% agarose gels. All PCR products were quantified by Quant-iT double-stranded DNA HS reagent and pooled. High-throughput sequencing analysis of bacterial and fungal rRNA genes was performed on the purified, pooled sample by use of the Illumina NovaSeq 6000 system at Biomarker Technologies Corporation (BMK), Beijing, China.

**Bioinformatics analysis.** The raw reads obtained were quality checked and filtered, and reads with a final length of <20 bases were discarded. The remaining sequences without ambiguous bases were assigned to different OTUs at 97% sequence identity by VSEARCH (v10.0). These sequences were determined using Silva (release 128; http://www.arb-silva.de) for bacteria and Unite (release 7.2; http://unite.ut.ee/index.php) for fungi to identity these OTUs. The obtained OTU table was used to determine taxonomic relative abundances and subsequent diversity analysis.

**Alpha and beta diversity analysis.** The alpha diversity indices that were calculated using Mothur software (v.1.30) included the Chao index (https://mothur.org/wiki/chao/), the Shannon index (https://mothur.org/wiki/shannon/), the ACE index (https://mothur.org/wiki/ace/), and the Simpson index (https://mothur.org/wiki/simpson/).

Principal-coordinate analysis (PCoA) and the statistically significant differences of bacterial or fungal communities among the three different parts of EI and EF seeds were tested through permutational multivariate analysis of variance (PERMANOVA) and analysis of similarity (ANOSIM) based on the Bray-Curtis dissimilarities using the vegan package in R (v.4.0.3).

**Metabolite extraction.** Twelve seed samples (six EI samples, six EF samples) were randomly selected from the 16 microbial sequencing samples for the detection of seed metabolites. Twelve glume samples (six EI samples, six EF samples) were randomly selected from the 16 microbial sequencing samples for the detection of glume metabolites. These EI/EF seed and glume samples were weighed into eppendorf (EP) tubes, respectively, and 500 $\mu$L extraction solution (acetonitrile-methanol-water, 2: 2: 1) containing an isotopically labeled internal standard mixture was added. After 30 s of vortexing, these samples were homogenized at 35 Hz for 4 min and sonicated for 5 min in an ice-water bath. The homogenization and sonication cycle was repeated 3 times. These samples were then stored at −40℃ for 1 h and then centrifuged at 12,000 rpm for 15 min at 4℃. Two hundred fifty microliters of supernatant was transferred to a fresh tube and dried in a vacuum concentrator at 37℃. The dried samples were reconstituted in 400 $\mu$L of 50% acetonitrile by sonication on ice for 10 min and then centrifuged at 13,000 rpm for 15 min at 4℃, and 75 $\mu$L of supernatant was transferred to a fresh glass vial for LC-MS analysis.

The ultra-high performance liquid chromatography (UHPLC) separation was carried out using an ExionLC Infinity series UHPLC system (AB Sciex), equipped with a UPLC BEH amide column (2.1 by 100 mm, 1.7 $\mu$m; Waters). The mobile phase consisted of 25 mmol/L ammonium acetate and 25 mmol/L ammonia hydroxide in water (pH 9.75) (A) and acetonitrile (B). The analysis was carried with an elution gradient as follows: ~0 to 0.5 min, 95% B; ~0.5 to 7.0 min, ~95% to 65% B; ~7.0 to 8.0 min, ~65% to 40% B; ~8.0 to 9.0 min, 40% B; ~9.0 to 9.1 min, ~40% to 95% B; ~9.1 to 12.0 min, 95% B. The column temperature was 25℃. The autosampler temperature was 4℃, and the injection volume was 2 $\mu$L (positive) or 2 $\mu$L (negative), respectively.

The TripleTOF 5600 mass spectrometer (AB Sciex) was used for its ability to acquire tandem MS (MS/MS) spectra on an information-dependent basis during an LC-MS experiment. In this mode, the acquisition software (Analyst TF 1.7; AB Sciex) continuously evaluates the full-scan survey MS data as it collects and triggers the acquisition of MS/MS spectra depending on preselected criteria. In each cycle, the most intensive 12 precursor ions with an intensity above 100 were chosen for MS/MS at a collision energy of 30 eV. The cycle time was 0.56 s. Electrospray ionization source conditions were set as follows: gas 1 as 60 lb/in$^2$, gas 2 as 60 lb/in$^2$, curtain gas as 35 lb/in$^2$, source temperature as 600℃, declustering potential as 60 V, and ion spray voltage floating as 5,000 V or −4,000 V in positive or negative mode, respectively. The identification of metabolites of seeds and glumes was done at Biomarker Technologies Corporation (BMK), Beijing, China.

**Metabolomics analysis.** The different variables were screened through principal-component analysis (PCA) and orthogonal partial least-squares discriminant analysis (OPLS-DA). Different metabolites were filtered according to variable importance in the project (VIP) of >1, $P$ value of <0.05, and fold change (FC) of >2. The KEGG database was used to analyze the pathway and enzyme for annotation, enrichment, and classification.

**Treatments in the study.** In this study, according to the infection status of the *E. gansuensis* endophyte and different parts of EI and EF seeds, the seed-borne microbial communities were defined as six groups. These were SEpI (seed epiphytic microbes of endophyte-infected plants), SEpF (seed epiphytic microbes of endophyte-free plants), SEnI (seed endophytic microbes of endophyte-infected plants), SEnF (seed endophytic microbes of endophyte-free plants), GEpI (glume microbial community of endophyte-infected plants), and GEpF (glume microbial community of endophyte-free plants).

According to the *E. gansuensis* infection status and the different parts of EI and EF seeds, the seed metabolites were defined as four groups. These were SEI (seed metabolites of endophyte-infected plants), SEF (seed metabolites of endophyte-free plants), GEI (glume metabolites of endophyte-infected plants), and GEF (glume metabolites of endophyte-free plants).

**Statistical analyses.** Visualizations were performed using SigmaPlot (v.12.5) and R software (v.4.0.3) packages (ggplot2 v.3.3.5; igraph v.1.2.6). Differences in seed-borne microbial community diversity under the infection status of the *E. gansuensis* endophyte and different parts of EI and EF seeds were tested using a two-way analysis of variance (two-way ANOVA) in SPSS 22.0. Significant differences in seed-borne microbial community diversity between EI and EF *A. inebrians* seeds with treatment of different parts of EI and EF seeds were examined by an independent-sample $t$ test, and significant differences in seed-borne microbial community diversity among the treatment of different parts of EI and EF seeds were examined by one-way analysis of variance (one-way ANOVA). In all tests, a $P$ value of <0.05 was considered a significant difference. For correlation network analysis, if the sample had both metabolite and microbial data, Spearman's correlation via R software (v.4.0.3) packages (stats v.3.3.5) was used for the sample (80, 81). Metabolite data and microbial data were paired based on individual seed samples to avoid the influence of sample diversity on correlation results.

**Compliance with ethics requirements.** This study did not involve any studies with human or animal subjects.

**Data availability.** The original sequence data of bacterial and fungal sequences were submitted to the NCBI database (https://www.ncbi.nlm.nih.gov/). The BioProject accession numbers of bacterial and fungal raw data are PRJNA865502 and PRJNA865530, respectively.

## SUPPLEMENTAL MATERIAL

Supplemental material is available online only.

**SUPPLEMENTAL FILE 1**, PDF file, 1.2 MB.

**SUPPLEMENTAL FILE 2**, XLSX file, 0.9 MB.

## ACKNOWLEDGMENTS

We thank the editor and anonymous reviewers for their valuable comments.

This work was financially supported by the National Nature Science Foundation of China (grant no. 32061123004 and 31772665), the National Basic Research Program of China (grant

no. 2014CB138702), and the Fundamental Research Funds for the Central Universities (grant no. jbky-2022-ey21) at Lanzhou University and the Key Scientific Research Platform and Project of Guangdong Education Department (grant no. 2021KCXTD054).

We declare no conflicts of interest.

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
