## [Reviewer comments · Microbiology Spectrum]

Microbiology Spectrum

The effect of *Epichloë gansuensis* endophyte on seed-borne microbes and seed metabolites in *Achnatherum inebrians*

Xingxu Zhang, Jinjin Liang, Guoyu Gao, Rui Zhong, Bowen Liu, Michael Michael J. Christensen, Yawen Ju, Wu Zhang, Yanzhong Li, Chunjie Li, and Zhibiao Nan

Corresponding Author(s): Xingxu Zhang, Lanzhou University

Review Timeline:

Submission Date:	April 27, 2022
Editorial Decision:	July 27, 2022
Revision Received:	August 25, 2022
Editorial Decision:	October 24, 2022
Revision Received:	November 22, 2022
Editorial Decision:	December 17, 2022
Revision Received:	January 3, 2023
Accepted:	January 23, 2023

Editor: Patricia Albuquerque

Reviewer(s): The reviewers have opted to remain anonymous.

Transaction Report:

DOI: <https://doi.org/10.1128/spectrum.01350-22>

July 27, 2022

Dr. Xingxu Zhang
Lanzhou University
Lanzhou
China

Re: Spectrum01350-22 (The effect of *Epichloë gansuensis* endophyte on seed-borne microbes and seed metabolites in *Achnatherum inebrians*)

Dear Dr. Xingxu Zhang:

Thank you for submitting your manuscript to Microbiology Spectrum. Your manuscript needs a general English revision and our major concern is the statistical analysis. We strongly suggest you follow the suggestions of both reviewers, especially regarding an improvement of the statistical analysis.

Link Not Available

Sincerely,

Patricia Albuquerque

Journals Department
Reviewer comments:

Reviewer #1 (Comments for the Author):

This manuscript explores how the presence of a dominant fungal symbiont in Drunken Horse grass affects the colonization of host plant seeds by other fungi and bacteria as well as the metabolites detected within seeds. The study was well designed where biological materials were collected from controlled experimental plots in the field and where powerful sequencing and analytical chemistry techniques were utilized. There are a lot of data presented on microbial communities and on the host plant metabolome, and there seems to be some clear patterns but see below.

My major concern is that the paper analyzes so many microbial and chemical variables, and calculates many hundreds of

correlations among these variables (see Fig. 4) that the issue of statistical tests and multiple comparisons arise. On line 363 nearly 900 individual correlations are mentioned, and then 189 more on 366 and more beyond that (see also line 385, 400, 416). I am not a statistician but I don't think that this large number of single correlations is the correct or appropriate way to present the data. I recommend that a statistician read the manuscript and comment on this issue. It is also very challenging to extract any general patterns from the multiple panels in Fig. 4 so a better way of condensing and presenting these data is necessary. Network analysis? Can microbes be grouped into higher categories or metabolites be grouped by larger categories? As it stands now, this reader was overwhelmed by the sheer number of comparisons and correlations given their sheer number. The text between 360 and 451 is hard to digest, and are the many panels in Fig. 4.

Minor comments

32 - LEp - what is it? See Line 52 as well.

37 - what kind of metabolites?

38 - 724 significant correlations out of how many possible? Does this take into account multiple comparisons given that 5% might be expected to be significant purely by chance? See 250-251 as well. I am not a statistician but I don't think that $P < .05$ is the appropriate criteria. See 363-365 as well.

36-43 - This large number of correlations and comparisons need to be put into a broader context.

62 - Is the name of the fungus uncertain, or are there two different fungi? 122-135 seems to focus on just one of the names.

92 - typo? Metabolomics?

99-107 - clarify whether metabolites are produced by the plant or by the microbe. See 131 as well. If metabolites produced by the endophyte then differences between EI and EF would be expected.

133 - this hypothesis could be more specific.

139 - were the two types of plants treated the same with respect to water, fertilizers, etc.?

148 - A composite sample of all plots, subplots within one plot? How many plots were EI vs. EF?

152 - Explain this washing more. Were epiphytes washed off, or was the wash sampled?

161 - What was the reason that glumes were examined, vs. leaves or stems? It is clear why seeds were examined but less clear about glumes.

240 - I don't understand this VIP

263 - what treatments? EI vs. EF?

Some grammar and punctuation throughout the manuscript needs to be carefully edited.

282 - I don't understand why there are 4 p-values vs. 1 or 2?

292 - location? What does this refer to?

300 and 302 - it seems odd that Shannon and Simpson are in opposite directions.

306 - I don't think that S, NS and F treatments were ever explained in the text. What are they? I used a search function and this is the first time these labels are used so they need to be defined. They are defined in a figure caption but not the text.

311 - 517 and 517?

325-326 - what do you mean by up and down-regulated? I don't think there was any transcriptome analyses. 334-335 suggests that you mean comparisons between EI and EF.

328 -330 - a wide variety of metabolites. Can you say which are microbial origin vs. plant origin?

359 - in this whole section it should be indicated somewhere that these correlations could arise from responses to some other third factor that was not measured in the study. For example, a lower pH could favor certain group of fungi and affect particular metabolites without a direct interaction between the two.

494-496 - can any of these postulated mechanisms be tested with your data?

506 - see comment about 152 as well.

513 - give scientific name.

527 - *Fusarium* is a very large genus and may not necessarily represent pathogens here.

528-531 - very speculative in the absence of supporting data.

553-554 - what other metabolites?

588 - mention this hypothesis again?

Reviewer #2 (Comments for the Author):

Summary:

What is the main message of the paper? This research study examined the microbiome of *Achnatherum inebrians* grass seeds infected (or not) with the *Epichloe gansuensis* endophyte. Both fungal and bacterial communities of seeds and glumes were determined. To determine the functionality of these microbial communities, metabolites of the seeds and glumes were assessed. The significance of this research lies in the invasiveness of the grass, a behavior that has a correlation to its infection by this fungal endophyte.

Are you convinced that the data presented supports the main conclusions?

The conclusion that *E. gansuensis* modulates the microbial diversity of DHG seeds and glumes is not supported. Rather, the conclusion that microbial diversity and endophyte infection are correlated is supported. The conclusion that the endophytic infection of DHG by *E. gansuensis* changed the metabolic profiles of both seeds and glumes is supported. Significant differences, both positive and negative, were found for microbiomes and metabolomics of the endophyte-infected grasses compared with those not infected.

Major points:

1. Since these are field-grown grasses, the analyses would benefit from the inclusion of additional metadata, including soil-borne microbiome data, age of the grass at the time of harvest, and time since endophyte infection.
2. Other studies showed bacteria associated with leaf tissues may be the source of seed bacteria, but the claim is made in this current research study that there is strong support that the presence of an *Epichloë* endophyte modifies the foliar bacterial microbial communities. I think there is some confusion as to cause and effect, which could be cleared up with further studies.
3. Figure S2 references a core microbiome, but this is not discussed, nor is the figure referenced, in the manuscript. The concept of a core microbiome is central to understanding cause and effect related to endophyte infection and changes to the microbiomes of the seeds and glumes.

Minor points:

1. Methodologically, how epiphytic contamination of glumes was prevented when the glumes were removed for analysis was not addressed.
2. In the results section (lines 254-277) it is unclear if endophyte-infected seeds are being discussed, or the totality of the microbiome. It is more apparent in the associated figures, but this section is confusing as written.
3. Figure S1 refers to glumes as 'L', but 'F' in Figure 1. Relative abundances in Tables S1 and S2 should be adjusted to represent percentages.
4. There are instances where the manuscript is too verbose, repetitive, and confusing. For instance, lines 85-107 could be written more clearly to present the significance of metabolomics to the research. The reporting of the ACE and Chao indices are also confusing (lines 279-282).
5. On lines 182 and 184, two different sequencing platforms are noted.

Staff Comments:

Preparing Revision Guidelines

Please return the manuscript within 60 days; if you cannot complete the modification within this time period, please contact me. If you do not wish to modify the manuscript and prefer to submit it to another journal, please notify me of your decision immediately so that the manuscript may be formally withdrawn from consideration by Microbiology Spectrum.

Reviewer #1:

This manuscript explores how the presence of a dominant fungal symbiont in Drunken Horse grass affects the colonization of host plant seeds by other fungi and bacteria as well as the metabolites detected within seeds. The study was well designed where biological materials were collected from controlled experimental plots in the field and where powerful sequencing and analytical chemistry techniques were utilized. There are a lot of data presented on microbial communities and on the host plant metabolome, and there seems to be some clear patterns but see below.

My major concern is that the paper analyzes so many microbial and chemical variables, and calculates many hundreds of correlations among these variables (see Fig. 4) that the issue of statistical tests and multiple comparisons arise. On line 363 nearly 900 individual correlations are mentioned, and then 189 more on 366 and more beyond that (see also line 385, 400, 416). I am not a statistician but I don't think that this large number of single correlations is the correct or appropriate way to present the data.

We sincerely thank you for the very thorough and informed revision of our manuscript and the important concerns and comments that you have made. We have carefully studied what you have indicated and have endeavored to make the improvements required.

Response: We chose this correlation analysis referred by Noecker et al. (2019), Fu et al. (2020) and Huang et al. (2022). The main reason for this correlation analysis procedure with a large number of individual correlations is that we divided seed-borne microbes into three parts and divided these metabolites into two parts for analysis, and the bacteria and fungi were treated separately for analysis, so there are six combinations: seed epiphytic bacteria and seed metabolites; seed endophytic bacteria and seed metabolites; glumes bacteria and glumes metabolites; seed epiphytic fungi and seed metabolites; seed endophytic fungi and seed metabolites; glumes fungi and glumes metabolites, so there were many treatments involved in this manuscript. And another important reason is the extensive screening criteria of differential metabolites in seeds and glumes; we have a large number of differential metabolites to analyze in correlations between differential metabolites and microbes in the initial manuscript. In order to correct this major point following your advice, we changed the screening criteria of differential metabolites of seed and glumes from “FC>1, VIP>1, P<0.05” to “FC>2, VIP>1, P<0.05”. Through the above series of operations, we

got a total of 170 significant correlations between seed-borne bacteria and metabolites of seed and glumes, and a total of 124 significant correlations between seed-borne fungi and metabolites of seed and glume. This revision has made the content simpler, Figure 4 clearer, and is now more comprehensible. Please check these revised Figure 4 and Table S5-12 data in revised manuscript.

References:

1. Noecker C, Chiua HC, McNallya CP, Borenstein E. Defining and evaluating microbial contributions to metabolite variation in microbiome-metabolome association studies. *mSystems*. 2019, 4(6): e00579-19.
2. Fu M, Zhang XW, Liang YH, Lin SR, Qian WP, Fan SR. Alterations in vaginal microbiota and associated metabolome in women with recurrent implantation failure. *mBio*. 2020, 11: e03242-19.
3. Huang K, Wang YG, Bai Y, Luo QY, Lin XC, Yang QY, Wang SH, Xin HJ. Gut microbiota and metabolites in atrial fibrillation patients and their changes after catheter ablation. *Microbiology Spectrum*. 2022, 10(2): e01077-21.

I recommend that a statistician read the manuscript and comment on this issue. It is also very challenging to extract any general patterns from the multiple panels in Fig. 4 so a better way of condensing and presenting these data is necessary. Network analysis?

Response: We changed the criterion for seed and glumes differential metabolites from “FC>1, VIP>1, P<0.05” to “FC>2, VIP>1, P<0.05”. Based on the previous criterion “FC>1, VIP>1, P<0.05”, we have got so many differential metabolites of seed and glumes; we feel that it is disadvantageous for us to show and explore the close relationships between microbe and metabolites. Figure 4 and Table S5-12 were redone to present these relationships. Please check the revision data in revised manuscript.

Microorganisms work together to form complicated biological networks through negative (predation), positive (mutualistic symbiosis) and neutral (commensalism) relationships (Faust and Raes, 2012). Deng et al. (2012) and Zhou et al. (2011) using RMT (random matrix theory) developed a powerful molecular ecological network analysis method to analysis network interactions in microbial communities. Network analysis have been successfully used to explore complex microbial communities in groundwater (Deng et al., 2012, 2016) and soil (Zhou et al., 2010, 2011; Wang et al., 2018; Zhang et al., 2018; Jiao et al., 2019), rhizosphere (Mendes et al., 2013; Hameed et al., 2015; Shi et al., 2016) and human gut (Ley et al., 2006; Faust et al., 2012).

Network analysis also successfully used in analyzing correlations between different metabolites (Mallott et al., 2022). For analyzing correlations between metabolites and microbes, network analysis isn't a particularly suitable analysis method.

Reference:

1. Faust K, Raes J. Microbial interactions: from networks to models. *Nature Reviews Microbiology*. 2012, 10(8): 538-550.
2. Deng Y, Jiang YH, Yang YF, He ZL, Luo F, Zhou JZ. Molecular ecological network analyses. *BMC Bioinformatics*. 2012, 13: 113.
3. Zhou JZ, Deng Y, Luo F, He ZL, Yan YF. Phylogenetic molecular ecological network of soil microbial communities in response to elevated CO₂. *mBio*. 2011, 2(4): e00122-11.
4. Deng Y, Zhang P, Qin YJ, Tu QC, Yang YF, He ZL, Warren SC. Network succession reveals the importance of competition in response to emulsified vegetable oil amendment for uranium bioremediation. *Environmental Microbiology*. 2016, 18(1): 205-218.
5. Zhou JZ, Deng Y, Luo F, He ZL, Tu QC, Zhi XY. Functional molecular ecological networks. *mBio*. 2010, 1(4): e00169-10.
6. Wang S, Wang XB, Han XG, Deng Y. Higher precipitation strengthens the microbial interactions in semi-arid grassland soils. *Global Ecology and Biogeography*. 2018, 27(5): 570-580.
7. Zhang BG, Zhang J, Liu Y, Shi P, Wei GH. Co-occurrence patterns of soybean rhizosphere microbiome at a continental scale. *Soil Biology & Biochemistry*. 2018, 118(1): 178-186.
8. Jiao S, Yang YF, Xu YQ, Zhang J, Lu YH. Balance between community assembly processes mediates species coexistence in agricultural soil microbiomes across eastern China. *The ISME Journal*. 2019, 14(1): 202-216.
9. Mendes R, Garbeva P, Raaijmakers JM. The rhizosphere microbiome: significance of plant beneficial, plant pathogenic, and human pathogenic microorganisms. *FEMS Microbiology Reviews*. 2013, 37(5): 634-663.
10. Hameed A, Yeh MW, Hsieh YT, Chung WC, Lo CT, Young LS. Diversity and functional characterization of bacterial endophytes dwelling in various rice (*Oryza sativa* L.) tissues, and their seed-borne dissemination into rhizosphere under gnotobiotic P-stress. *Plant and Soil*. 2015, 394(1-2): 177-197.
11. Shi SJ, Nuccio EE, Shi ZJ, He ZL, Zhou JZ, Firestone MK. The interconnected rhizosphere:

High network complexity dominates rhizosphere assemblages. *Ecology Letters*. 2016, 19(8): 926-936.

12. Ley RE, Turnbaugh PJ, Klein S, Gordon JI. Microbial ecology human gut microbes associated with obesity. *Nature*. 2006, 444(7122): 1022-1023.

13. Faust K, Sathirapongsasuti JF, Izard J, Segata N, Gevers D, Raes J, Huttenhower C. Microbial co-occurrence relationships in the human microbiome. *PLoS Computational Biology*. 2012, 8(7): e1002606.

14. Mallott EK, Skovmand LH, Garber PA, Amato KR. The faecal metabolome of black howler monkeys (*Alouatta pigra*) varies in response to seasonal dietary changes. *Molecular Ecology*, 2022, 31: 4146-4161.

Can microbes be grouped into higher categories or metabolites be grouped by larger categories?

Response: We grouped these differential seed and glumes metabolites into a larger group, and now these metabolites are in the primary group, and before that they were in the secondary group; the classified criterion was according to the HMDB data base (Wishart et al., 2007, 2009). Please check the Table 2 in revised manuscript.

Reference:

1. Wishart DS, Dan T, Knox C, et al. HMDB: The human metabolome database. *Nucleic Acids Research*. 2007, 35: D521-526.

2. Wishart DS, Craig K, Chi GA, et al. HMDB: a knowledgebase for the human metabolome. *Nucleic Acids Research*. 2009, 37: 603-610.

As it stands now, this reader was overwhelmed by the sheer number of comparisons and correlations given their sheer number. The text between 360 and 451 is hard to digest, and are the many panels in Fig. 4.

Response: Based on the revision of Figure 4 and Table S5-12, we revised and simplified the content of this part to make it clearer and more convenient for readers. Please check the revision in our revised manuscript (Line 380-440).

Minor comments

32 - LEp - what is it? See Line 52 as well.

Response: LEp was the glumes with endophyte-infected, the LEp should be changed to FEI (glumes of endophyte-infected seeds). We have corrected this mistake and added explanation of

FEI according to your comments in our revised manuscript. Please check it in revised manuscript (Line 32).

37 - what kind of metabolites?

Response: Based on the FC (fold change) >2 , VIP >1 and P <0.05 , the 108 differential metabolites between EI and EF seeds were separated, and 31 differential metabolites between EI and EF glumes were separated.

38 - 724 significant correlations out of how many possible? Does this take into account multiple comparisons given that 5% might be expected to be significant purely by chance? See 250-251 as well. I am not a statistician but I don't think that P <0.05 , the appropriate criteria. See 363-365 as well.

Response: There were total 724 significant correlations among metabolites and microbes. We analyzed the correlation by the Spearman's rank method, without taking into account multiple comparisons. This method was referenced by Noecker et al. (2019), Fu et al. (2020) and Huang et al. (2022). We have addressed the problem that correlation results are too many to be trusted from the source of correlation analysis, and we have changed the selection criteria of seed and glume differential metabolites from "FC >1 , VIP >1 , P <0.05 " to "FC >2 , VIP >1 , P <0.05 ". Based on the above series of operations, we got a total of 170 significant correlations between seed-borne bacteria and metabolites of seed and glumes, and a total of 124 significant correlations between seed-borne fungi and metabolites of seed and glume.

Reference:

1. Noecker C, Chiua HC, McNallya CP, Borenstein E. Defining and evaluating microbial contributions to metabolite variation in microbiome-metabolome association studies. *mSystems*. 2019, 4(6): e00579-19.
2. Fu M, Zhang XW, Liang YH, Lin SR, Qian WP, Fan SR. Alterations in vaginal microbiota and associated metabolome in women with recurrent implantation failure. *mBio*. 2020, 11: e03242-19.
3. Huang K, Wang YG, Bai Y, Luo QY, Lin XC, Yang QY, Wang SH, Xin HJ. Gut microbiota and metabolites in atrial fibrillation patients and their changes after catheter ablation. *Microbiology Spectrum*. 2022, 10(2): e01077-21.

36-43 - This large number of correlations and comparisons need to be put into a broader context.

Response: After we changed the criterion for seed and glumes differential metabolites from

“FC>1, VIP>1, P<0.05” to “FC>2, VIP>1, P<0.05”; there were lower number of significant correlations between microbes and metabolites than in the initial manuscript. Please check the revision in revised manuscript.

62 - Is the name of the fungus uncertain, or are there two different fungi? 122-135 seems to focus on just one of the names.

Response: These are two *Epichloë* endophyte species that infect *Achnatherum inebrians*. Researchers isolated and identified *Epichloë gansuensis* and *E. inebrians* from Gansu and Xinjiang province in China, respectively. So almost every *A. inebrians* plant was found in surveys to be infected by either *E. gansuensis* (Li et al., 2004; Leuchtmann et al., 2014) or *E. inebrians* (Chen et al., 2015). In recent years, our research team has used seeds originating from those in the Li et al. (2004) study for relevant research work (Zhong et al., 2018, Ju et al., 2020; Liu et al., 2022) and in the current study.

References:

1. Li CJ, Nan ZB, Paul VH, Dapprich PD, Liu Y. A new *Neotyphodium* species symbiotic with drunken horse grass (*Achnatherum inebrians*) in China. *Mycotaxon*. 2004, 90: 141-147.
2. Leuchtmann, A., Bacon, C.W., Schardl, C.L., White, J.F., and Tadych, M. Nomenclatural realignment of *Neotyphodium* speices with genus *Epichloë*. *Mycologia*. 2014, 106(2): 202-215.
3. Chen L, Li XZ, Li CJ, Swoboda GA, Young CA, Sugawara K, Leuchtmann A, Schardl CL. Two distinct *Epichloë* species symbiotic with *Achnatherum inebrians*, drunken horse grass. *Mycologia*. 2015, 107: 863-873.
4. Zhong R, Xia C, Ju YW, Li NN, Zhang XX, Nan ZB, Christensen MJ. Effects of *Epichloë gansuensis* on root-associated fungal communities of *Achnatherum inebrians* under different growth conditions. *Fungal Ecology*. 2018, 31: 29-36.
5. Ju YW, Zhong R, Christensen MJ, Zhang XX. Effects of *Epichloë gansuensis* endophyte on the root and rhizosphere soil bacteria of *Achnatherum inebrians* under different moisture conditions. *Frontiers in Microbiology*. 2020, 11: 747.
6. Liu BW, Ju YW, Xia C, Zhong R, Christensen MJ, Zhang XX, Nan ZB. The effect of *Epichloë* endophyte on phyllosphere microbes and leaf metabolites in *Achnatherum inebrians*. *iScience*. 2022, 25: 104144.

92 - typo? Metabolomics?

Response: We have corrected this mistake in our revised manuscript.

99-107 - clarify whether metabolites are produced by the plant or by the microbe. See 131 as well.

If metabolites produced by the endophyte then differences between EI and EF would be expected.

Response: These metabolites in this manuscript were produced by DHG with *Epichloë* endophyte-infected (EI) or endophyte-free (EF); it is hard to identify whether they were produced by the plant or by the microbes. Metabolites different between the EI and EF DHG were alkaloids including neurotropic lysergic acid amides (ergot alkaloids) and paxilline (an indole-diterpene alkaloid) (Chen et al., 2015). But these alkaloids were not detected in these 517 seeds metabolites or 517 glumes metabolites. There were differential metabolites apart from these alkaloids between EI and EF; the detailed data can be found in Table S3-S4.

Reference:

1. Chen L, Li XZ, Li CJ, Swoboda GA, Schardl CL. Two distinct *Epichloë* species symbiotic with *Achnatherum inebrians*, drunken horse grass. *Mycologia*. 2015, 107(4): 863-873.

133 - this hypothesis could be more specific.

Response: We have corrected this according to your suggestion. Please check it in revision manuscript (Line 128-130).

139 - were the two types of plants treated the same with respect to water, fertilizers, etc.?

Response: The two types of plants received the same treatments including water, fertilizers and field management referred Liu et al. (2022).

Reference:

1. Liu BW, Ju YW, Xia C, Zhong R, Christensen MJ, Zhang XX, Nan ZB. The effect of *Epichloë* endophyte on phyllosphere microbes and leaf metabolites in *Achnatherum inebrians*. *iScience*, 2022, 25: 104144.

148 - A composite sample of all plots, subplots within one plot? How many plots were EI vs. EF?

Response: A total of 12 plots (each 4.8 x 4.0 m) were established in 2017 in this experimental field. Each plot was divided into two sub-plots by a 1 m deep concrete wall. One sub-plot of each plot was planted with endophyte-infected (EI) plants, and the other sub-plot was planted with endophyte-free (EF) plants. Samples from eight plots were randomly selected. EI and EF samples were obtained from two sub-plots of each plot. A total of eight plots were EI and eight plots were EF (Liu et al., 2022).

Reference:

1. Liu BW, Ju YW, Xia C, Zhong R, Christensen MJ, Zhang XX, Nan ZB. The effect of *Epichloë* endophyte on phyllosphere microbes and leaf metabolites in *Achnatherum inebrians*. *iScience*, 2022, 25: 104144.

152 - Explain this washing more. Were epiphytes washed off, or was the wash sampled?

Response: Thanks for raising this concern. Epiphytic microbes were washed from seed surfaces. 50 seeds were transferred into 50 mL plastic tubes filled with 30-40 mL PBS buffer, along with two blank controls without added seeds, followed by oscillation for 30-60 min at 150-200 r/min, sonication for 5 min, and further oscillation for 30-60 min at 150-200 r/min. The seeds were removed and the suspension was centrifuged at 10,000 g for 10 min to obtain precipitates containing bacteria, fungal spores and hyphae dislodged from the surface of seeds. These precipitates were used to detect the epiphytic microbes. This method was referred by (Links et al., 2014; Hill et al., 2005).

Reference:

1. Links MG, Demekke T, Grafenhan T, Hill JE, Hemmingsen SM, Hemmingsen SM. Simultaneous profiling of seed-associated bacteria and fungi reveals antagonistic interactions between microorganisms within a shared epiphytic microbiome on *Triticum* and *Brassica* seeds. *New Phytologist*, 2014, 202(2): 542-553.

2. Hill JE, Hemmingsen SM, Goldade BG, Dumonceaux TJ, Klassen J, Zijlstra RT, Goh SH, Van Kessel AG. 2005. Comparison of ileum microflora of pigs fed corn-, wheat-, or barley-based diets by chaperonin-60 sequencing and quantitative PCR. *Applied and Environmental Microbiology*, 2005, 71: 867-875.

161 - What was the reason that glumes were examined, vs. leaves or stems? It is clear why seeds were examined but less clear about glumes.

Response: Seed-borne microbial communities can have significant effects on plant health (McKellar and Nelson, 2003) and on longer-term seedling establishment (Morpeth and Hall, 2000; Nelson, 2018). In addition, studies evaluated that glumes also had an essential role in seed germination and seedling vigour (Raviv et al., 2018), even in grain production (Amarasinghe et al., 2020; Xie et al., 2022). The aim of this study is to investigate the effects of *Epichloë gansusensis* endophyte on seed-borne microbes and seed metabolites in *Achnatherum inebrians*, taking into

account the seeds and the seed accessory structure, glumes.

Reference:

1. McKellar ME, Nelson EB. Compost-induced suppression of *Pythium* damping-off is mediated by fatty-acid-metabolizing seed-colonizing microbial communities. *Applied and Environmental Microbiology*. 2003, 69: 452-460.
2. Morpeth DR, Hall AM. Microbial enhancement of seed germination in *Rosa corymbifera* 'Laxa'. *Seed Science Research*. 2000, 10: 489-494.
3. Nelson EB. The seed microbiome: Origins, interactions, and impacts. *Plant and Soil*. 2018, 422: 7-34.
4. Raviv B, Godwin J, Granot G, Grafi G. The dead can nurture: novel insights into the function of dead organs enclosing embryos. *International Journal of Molecular Sciences*. 2018, 19: 2455.
5. Amarasinghe YPJ, Kuwata R, Nishimura A, et al. Evaluation of domestication loci associated with awnlessness in cultivated rice, *Oryza sativa*. *Rice*. 2020, 13: 26.
6. Xie P, Tang SY, Chen CX, et al. Natural variation in glume coverage 1 causes naked grains in sorghum. *Nature Communications*. 2022, 13(1): 1068.

240 - I don't understand this VIP

Response: The VIP value (variable importance in the project) is the VIP value of OPLS-DA (orthogonal projections to latent structures- discriminant analysis, OPLS-DA) model (Figure S4 and S5) (Thevenot et al., 2015; Jia et al., 2019; Thevenot, 2022). We used the R (3.3.2) package *ropls* for OPLS-DA model calculation. The VIP value represents the influence of the difference between the corresponding metabolites in the classification of each group of samples in the model. It is generally believed that the metabolites with $VIP > 1$ are significantly different (Bi et al., 2022).

Reference:

1. Thevenot EA, Roux A, Xu Y, et al. Analysis of the human adult urinary metabolome variations with age, body mass index, and gender by implementing a comprehensive workflow for univariate and OPLS statistical analyses. *Journal of Proteome Research*. 2015, 14(8): 3322-3335.
2. Jia GL, Sha K, Feng XD, Liu HJ. Post-thawing metabolite profile and amino acid oxidation of thawed pork tenderloin by HVEF-A short communication. *Food Chemistry*. 2019, 291: 16-21.
3. Thevenot EA. PCA, PLS(-DA) and OPLS(-DA) for multivariate analysis and feature selection of omics data. 2022. DOI: 10.18129/B9.bioc.ropls.

4. Bi WW, Zhao GX, Zhou YT, et al. Metabonomics analysis of flavonoids in seeds and sprouts of two Chinese soybean cultivars. Science Reports. 2022, 12: 5541.

263 - what treatments? EI vs. EF?

Response: Treatments were referred to three different parts of seeds, NS (seed endophyte), S (seed epiphyte) and F (glume). We have corrected this mistake according to your comments in our revised manuscript (Line 282).

Some grammar and punctuation throughout the manuscript needs to be carefully edited.

Response: Although we tried to ensure a high standard of writing, clearly some problems were present; we have checked the manuscript carefully and have corrected many mistakes.

282 - I don't understand why there are 4 p-values vs. 1 or 2?

Response: This sentence mainly described the treatment of three different parts of seeds and also had a significant influence on ACE index, Chao index, Shannon index and Simpson indexes. We have corrected this mistake according to your comments in our revised manuscript. Thanks for you again (Line 302-303).

292 - location? What does this refer to?

Response: The meaning of location was the treatment of three different parts of seeds. We have corrected this mistake according to your comments in our revised manuscript (Line 311).

300 and 302 - it seems odd that Shannon and Simpson are in opposite directions.

Response: In this study, the larger the Shannon index the higher was the diversity of a community, but with the Simpson index a higher value represents the lower diversity of a community (Hill et al., 2003; Grice et al., 2009). These computational formulas of the Shannon and Simpson in this study were as follows.

The Shannon index (H') was calculated using the following formula (<https://mothur.org/wiki/shannon/>):

$$H_{\text{Shannon}} = - \sum_{i=1}^{\text{Sobs}} \frac{n_i}{N} \ln \frac{n_i}{N}$$

Sobs=the number of observed OTUs

n_i =the number of individuals in OUT i

N=the total number of individuals in the community

The Simpson index was calculated using the following formula

(<https://mothur.org/wiki/simpson/>):

$$D_{\text{Simpson}} = \frac{\sum_{i=1}^{\text{Sobs}} ni(ni - 1)}{N(N - 1)}$$

Sobs=the number of observed OTUs

n_i=the number of individuals in OUT *i*

N=the total number of individuals in the community

Reference:

1. Hill TCJ, Walsh KA, Harris JA, Moffett BF. Using ecological diversity measures with bacterial communities. *FEMS Microbiology Ecology*. 2003, 43: 1-11.

2. Grice EA, Kong HH, Conlan S, et al. Topographical and temporal diversity of the human skin microbiome. *Science*. 2009, 324(5931): 1190-1192.

306 - I don't think that S, NS and F treatments were ever explained in the text. What are they? I used a search function and this is the first time these labels are used so they need to be defined. They are defined in a figure caption but not the text.

Response: Thanks for pointing out this problem. The treatment of three different parts of seeds, S, NS and F treatments are explained in materials and methods of the revised manuscript. Please check it in revised manuscript (Line 168-176).

311 - 517 and 517?

Response: A total of 517 detected metabolites were annotated in endophyte-infected (EI) and endophyte-free (EF) seeds of *A. inebrians*, and a total of 517 detected metabolites were annotated in endophyte-infected (EI) and endophyte-free (EF) glumes of *A. inebrians*.

325-326 - what do you mean by up and down-regulated? I don't think there was any transcriptome analyses.

Response: Based on the FC (fold change)>2, VIP (VIP value of OPLS-DA model)>1 and P<0.05, 108 and 31 differential metabolites between EI vs EF seeds and glumes were separated, respectively (Table S5). Among these 108 metabolites, there were 69 upregulated differential metabolites and 39 downregulated differential metabolites (Table S5). And there were 25 upregulated and 6 downregulated differential metabolites in glume (Table S6). In our present study, we haven't use transcriptome analyses. Upward change for differential metabolites was named upregulated, and the downward change of differential metabolites was named downregulated,

these upregulated and downregulated were referred in Zhang et al. (2021) and Fu et al. (2020).

Reference:

1. Zhang J, Feng D, Law HKW, Wu Y, Zhu GH, Huang WY, Kang Y. Integrative analysis of gut microbiota and fecal metabolites in rats after prednisone treatment. *Microbiology Spectrum*. 2021, 9: e00650-21.

2. Fu M, Zhang XW, Liang YH, Lin SR, Qian WP, Fan SR. Alterations in vaginal microbiota and associated metabolome in women with recurrent implantation failure. *mBio*. 2020, 11: e03242-19.

334-335 suggests that you mean comparisons between EI and EF.

Response: Thanks for this point. We added mean comparisons between EI and EF seeds, or EI and EF glumes. Please check it in revised manuscript (Line 352-356).

328 -330 - a wide variety of metabolites. Can you say which are microbial origin vs. plant origin?

Response: Alkaloids are the main differences in metabolites between EI and EF grasses; the endophyte-grass symbionts can produce four main kinds of important alkaloids, including indole-diterpene, pyrrolopyrazine, ergot alkaloids, and pyrrolizidine, which are not detected in endophyte-free plants (Schardl et al., 2013; Young et al., 2015). Four main kinds of alkaloids were significantly produced in many grasses such as *Lolium perenne* (Nicholson et al., 2015), *Festuca pratensis* (Vikuk et al., 2019), *Lolium arundinaceum* (Klotz, 2015) and *Achnatherum inebrians* (Zhang et al., 2014) infected with *Epichloë* endophyte. Importantly, the content of alkaloids produced by *Epichloë* endophytic fungi under in medium culture conditions was 35 or 37 times higher than that detected in the leaves of DHG plants infected by *Epichloë* endophyte (Li, 2005; Gao, 2006). Alkaloids detected from *Achnatherum inebrians* with *Epichloë* endophyte-infected mainly reported as ergonovine, lysergic acid amides (ergine), and ergonovinine, which were major ergot alkaloids; these alkaloids can't be detected in *A. inebrians* with endophyte-free (Zhang et al., 2014; Liang et al., 2017). Factors affecting the content of alkaloids produced by DHG mainly focused on cutting frequency and height (Zhang et al., 2011a) and salt and drought stress (Zhang et al., 2011b). Study found that cytotoxic effect of ergot alkaloids in *Achnatherum inebrians* infected by the *Epichloë gansuense* endophyte (Zhang et al., 2014).

There were high number of studies concentrated on the kind and content of alkaloids produced by EI grass but less concentrated on more than alkaloids. It is hard to determine or distinguish these metabolites produced by microbe or plants. In this study, we found that the presence of *E.*

gansuensis changed the kind and content of metabolites produced by *A. inebrians* seeds. So further research is required to study which metabolites are produced by microbes or by plants.

Reference:

1. Schardl CL, Young CA, Hesse U, et al. Plant-symbiotic fungi as chemical engineers: multi-genome analysis of the clavicipitaceae reveals dynamics of alkaloid loci. *PLoS Genetics*. 2013, 9(2): e1003323.
2. Young CA, Schardl CL, Panaccione DG, Florea S, Takach JE, Charlton ND, Moore N, Webb JS, Jaromczyk J. Genetics, genomics and evolution of ergot alkaloid diversity. *Toxins*. 2015, 7: 1273-1302.
3. Nicholson MJ, Eaton CJ, Stärkel C, Tapper BA, Cox MP, Scott B. Molecular cloning and functional analysis of gene clusters for the biosynthesis of indole-diterpenes in *Penicillium crustosum* and *P. janthinellum*. *Toxins*. 2015, 7: 2701-2722.
4. Vikuk V, Young CA, Lee ST et al. Infection rates and alkaloid patterns of different grass species with systemic *Epichloë* endophytes. *Applied and Environmental Microbiology*. 2019, 85(17): e00465-e519.
5. Klotz JL. Activities and effects of ergot alkaloids on livestock physiology and production. *Toxins*. 2015, 7: 2801-2821.
6. Li CJ. Biological and Ecological Characteristics of *Achnatherum inebrians*/*Neotyphodium* endophyte symbiont. Lanzhou University, Lanzhou. (In Chinese with English abstract).
7. Gao JH. Derection of ergot alkaloid production by *Neotyphodium gansuensis* in two and sequencing the *dmaW* gene of *N. lolii* isolates AR1 and Lp19. Lanzhou University, 2006, Lanzhou. (In Chinese with English abstract).
8. Zhang XX, Nan ZB, Li CJ, Gao K. Cytotoxic effect of ergot alkaloids in *Achnatherum inebrians* infected by the *Neotyphodium gansuense* endophyte. *Journal of Agricultural and Food Chemistry*. 2014, 62(30): 7419-7422.
9. Liang Y, Wang HC, Li CJ, Nan ZB, Li FD. Effects of feeding drunken horse grass infected with *Epichloë gansuensis*, endophyte on animal performance, clinical symptoms and physiological parameters in sheep. *BMC Veterinary Research*. 2017, 13: 223.
10. Zhang XX, Li CJ, Nan ZB. Effects of cutting frequency and height on alkaloid production in endophyte-infected drunken horse grass (*Achnatherum inebrians*). *Science China Life Sciences*.

2011a, 54(6): 567-571.

11. Zhang XX, Li CJ, Nan ZB. Effects of salt and drought stress on alkaloid production in endophyte infected drunken horse grass (*Achnatherum inebrians*). *Biochemical Systematics and Ecology*. 2011b, 39: 471-476.

359 - in this whole section it should be indicated somewhere that these correlations could arise from responses to some other third factor that was not measured in the study. For example, a lower pH could favor certain group of fungi and affect particular metabolites without a direct interaction between the two.

Response: Since our experiment concentrated on microorganisms and metabolites of seeds, and we did correlation analysis between microorganisms and metabolites in this process. Indeed, there is a lack of consideration of environmental factors, and we will carry out extensive related research in the subsequent work. These correlations between metabolites and microbes were reworked in this manuscript; please check it in revised manuscript.

494-496 - can any of these postulated mechanisms be tested with your data?

Response: We have corrected this sentence according to your suggestion. Please check it in revised manuscript (Line 480-483).

506 - see comment about 152 as well.

Response: Thanks for your suggestion. Epiphytic micorbiota were detected as following; a 10 g sample of each seed lot was soaked in a solution of 45 mL buffered peptone water (10 g peptone, 5 g NaCl, 3.5 g Na₂HPO₄, 1.5 g KH₂PO₄ l⁻¹) containing 0.05% Triton X-100 (Sigma) in a 250 mL Erlenmeyer flask at room temperature with shaking (150 rpm) for 1 h. The liquid fractions were centrifuged at 4000 g for 15 min and the supernatant discarded. Pellets were resuspended in 200 µL of TE buffer and subjected to DNA extraction using the previously described bead-beating protocol (Links et al., 2014). We have corrected this sentence according to your suggestion. Please check it in revised manuscript (Line 493-495).

Reference:

1. Links MG, Demeke T, Gräfenhan T, Hill JE, Hemmingsen SM, Dumonceaux TJ. 2014. Simultaneous profiling of seed-associated bacteria and fungi reveals antagonistic interactions between microorganisms within a shared epiphytic microbiome on *Triticum* and *Brassica* seeds. *New Phytologist*. 2014, 202(2): 542-553.

513 - give scientific name.

Response: We have corrected this mistake in our revised manuscript (Line 501).

527 - *Fusarium* is a very large genus and may not necessarily represent pathogens here.

Response: We have revised this in line 517: Of importance is that many *Fusarium* species are pathogens of a wide range of plants (Line 517-518).

528-531 - very speculative in the absence of supporting data.

Response: We have corrected this sentence according to your suggestion. Please check it in revised manuscript (Line 518-521).

553-554 - what other metabolites?

Response: Other metabolites include epichlicin and cyclosporin T which are reported in Song et al. (2020). We added this information in the revised manuscript. Please check it in revised manuscript (Line 544).

Reference:

1. Song, Q.Y., Li, F., Nan, Z.B., Coulter, J.A., and Wei, W.J. Do *Epichloë* endophytes and their grass symbiosis only produce toxic alkaloids to insects and livestock? *Journal of Agricultural and Food Chemistry*. 2020, 68: 1169-1185.

588 - mention this hypothesis again?

Response: We have added this hypothesis in revised manuscript. Please check the revision in revised manuscript (Line 577-579).

Thanks again for your good suggestions and comments!

Reviewer #2 (Comments for the Author):

Summary:

What is the main message of the paper? This research study examined the microbiome of *Achnatherum inebrians* grass seeds infected (or not) with the *Epichloë gansuensis* endophyte. Both fungal and bacterial communities of seeds and glumes were determined. To determine the functionality of these microbial communities, metabolites of the seeds and glumes were assessed. The significance of this research lies in the invasiveness of the grass, a behavior that has a correlation to its infection by this fungal endophyte. Are you convinced that the data presented supports the main conclusions? The conclusion that *E. gansuensis* modulates the microbial diversity of DHG seeds and glumes is not supported. Rather, the conclusion that microbial diversity and endophyte infection are correlated is supported. The conclusion that the endophytic infection of DHG by *E. gansuensis* changed the metabolic profiles of both seeds and glumes is supported. Significant differences, both positive and negative, were found for microbiomes and metabolomics of the endophyte-infected grasses compared with those not infected.

Response: Thanks for your good suggestions. *Epichloë gansuensis* endophyte modulation of the microbial diversity of DHG seeds and glumes were supported by alpha diversity and beta diversity; this viewpoint could be obtained from Figure 2, Figure 3 and Table 1. From Figure 2, we can find that the *E. gansuensis* endophyte significantly ($P < 0.05$) changed the ACE and Chao index of bacterial communities, and significantly ($P < 0.05$) changed the Simpson index of fungal communities. From Figure 3 and Table 1, we can found that *E. gansuensis* endophyte significantly ($P < 0.05$) changed the beta diversity of bacterial and fungal communities.

Major points:

1. Since these are field-grown grasses, the analyses would benefit from the inclusion of additional metadata, including soil-borne microbiome data, age of the grass at the time of harvest, and time since endophyte infection.

Response: Thanks for your good suggestions. These soil-borne microbiome data have not been measured yet. Phyllosphere microbe data have been successfully published in iScience (Liu et al., 2022). Seeds samples were collected from field growing plants in September 2019, these plots were established in 2017 in this experimental field (Liu et al., 2022). *Epichloë* endophytic fungus was grown from a seed after germination. The majority of *Epichloë* species are asexual

(*Neotyphodium* sp.) and vertically transmit their progeny through seeds (Schardl et al. 2004; Ekanayake et al. 2012). In our study, the EI and EF seeds used in this study originated from the EI and EF plants (Liu et al., 2022).

1. Liu BW, Ju YW, Xia C, Zhong R, Christensen MJ, Zhang XX, Nan ZB. The effect of *Epichloë* endophyte on phyllosphere microbes and leaf metabolites in *Achnatherum inebrians*. *iScience*. 2022, 25: 104144.

2. Schardl CL, Leuchtmann A, Spiering MJ. Symbioses of grasses with seedborne fungal endophytes. *Annual Review Plant Biology*. 2004, 55: 315-340.

3. Ekanayake NP, Hand ML, Spangenberg GC, Forster JW, Guthridge MK. Genetic diversity and host specificity of fungal endophyte taxa in fescue pasture grasses. *Crop Science*. 2012, 52: 2243-2252.

2. Other studies showed bacteria associated with leaf tissues may be the source of seed bacteria, but the claim is made in this current research study that there is strong support that the presence of an *Epichloë* endophyte modifies the foliar bacterial microbial communities. I think there is some confusion as to cause and effect, which could be cleared up with further studies.

Response: Thanks for your good comment. Many studies have found that phyllosphere microorganisms may be the source of seed-borne microorganisms (Compant et al., 2011; Truyens et al., 2015; Zhu et al., 2021). There were many studies that illustrated that *Epichloë* endophytic fungi had an effect on underground parts such as rhizosphere and root microbial diversity (Buyer et al., 2011; Casas et al., 2011; Roberts and Ferraro, 2015; Ju et al., 2020; Zhong et al., 2021). Our team's recent research has found that *Epichloë* endophytic fungi change the diversity of phyllosphere microbes and leaves metabolites (Liu et al., 2022), and our current research is about the effects of *Epichloë* endophytic fungi on seed-borne microbes and seed metabolites. Whether *Epichloë* endophytic fungi infection affects seed-borne microbes by changing phyllosphere microbes needs to be further studied.

Reference:

1. Compant S, Mitter B, Coli-Mull JG, Gangl H, Sessitsch A. Endophytes of grapevine flowers, berries, and seeds: identification of cultivable bacteria, comparison with other plant parts, and visualization of niches of colonization. *Microbial Ecology*. 2011, 62: 188-197.

2. Truyens S, Weyens N, Cuypers A, Vangronsveld J. Bacterial seed endophytes: genera, vertical

transmission and interaction with plants. *Environmental Microbiology Reports*. 2015, 7(1): 40-50.

3. Zhu YG, Xiong C, Wei Z et al. Impacts of global change on the phyllosphere microbiome. *New Phytologist*. 2021, 234: 1977-1986.

4. Buyer JS, Zuberer DA, Nichols KA, Franzluebbers AJ. Soil microbial community function, structure, and glomalin in response to tall fescue endophyte infection. *Plant and Soil*. 2011, 339: 401-412.

5. Casas C, Omacini M, Montecchia MS, Correa OS. Soil microbial community responses to the fungal endophyte *Neotyphodium* in Italian ryegrass. *Plant and Soil*. 2011, 340: 347-355.

4. Roberts EL, Ferraro A. Rhizosphere microbiome selection by *Epichloë* endophytes of *Festuca arundinacea*. *Plant and Soil*. 2015, 396: 229-239.

6. Ju YW, Zhong R, Christensen MJ, Zhang XX. Effects of *Epichloë gansuensis* endophyte on the root and rhizosphere soil bacteria of *Achnatherum inebrians* under different moisture conditions. *Frontiers in Microbiology*. 2020, 11: 747.

7. Zhong R, Xia C, Ju YW, Zhang XX, Duan TY, Nan ZB, Li CJ. A foliar *Epichloë* endophyte and soil moisture modified belowground arbuscular mycorrhizal fungal biodiversity associated with *Achnatherum inebrians*. *Plant and Soil*. 2021, 458(1-2): 105-122.

8. Liu BW, Ju YW, Xia C, Zhong R, Christensen MJ, Zhang XX, Nan ZB. The effect of *Epichloë* endophyte on phyllosphere microbes and leaf metabolites in *Achnatherum inebrians*. *iScience*. 2022, 25: 104144.

3. Figure S2 references a core microbiome, but this is not discussed, nor is the figure referenced, in the manuscript. The concept of a core microbiome is central to understanding cause and effect related to endophyte infection and changes to the microbiomes of the seeds and glumes.

Response: Thanks for your good comment, sorry for our mistake. We have corrected this mistake according to your comments in our revised manuscript. We have added a description of Figure S2 in the revised manuscript (Line 275-278).

Minor points:

1. Methodologically, how epiphytic contamination of glumes was prevented when the glumes were removed for analysis was not addressed.

Response: We wore sterile gloves and mask and separated glumes from seeds on an ultra-clean work table sterilized with ultraviolet light for 1 hour to ensure the presence of sterility throughout

separated process. These glumes removed from the seeds was immediately stored in sterile tubes, frozen in liquid nitrogen and then stored in -80°C refrigerator for subsequent sequencing (Links et al., 2014; Bastías et al., 2022). We have added these important points in the revised manuscript (Line 145-149).

Reference:

1. Links MG, Demeke T, Grafenhan T, Hill JE, Hemmingsen SM, Hemmingsen SM. Simultaneous profiling of seed-associated bacteria and fungi reveals antagonistic interactions between microorganisms within a shared epiphytic microbiome on *Triticum* and *Brassica* seeds. *New Phytologist*. 2014, 202(2): 542-553.

2. Bastías DA, Bustos LB, Jáuregui R, Barrera A, Acuña-Rodríguez IS, Molina-Montenegro MA Gundel PE. *Epichloë* fungal endophytes influence seed-associated bacterial communities. *Frontiers in Microbiology*. 2022, 12: 795354.

2. In the results section (lines 254-277) it is unclear if endophyte-infected seeds are being discussed, or the totality of the microbiome. It is more apparent in the associated figures, but this section is confusing as written.

Response: Sorry for our unclear thinking. We have rethought and reorganized this part. We analyzed and described the seed-borne microbes as a whole. Six different treatments (SEI, SEF, NSEI, NSEF, FEI and FEF) of seed-borne microbe as an important point, microbial composition of these treatments were introduced in detail. We have added many important points in this part. Please check the revision in revised manuscript (Line 270-296).

3. Figure S1 refers to glumes as 'L', but 'F' in Figure 1. Relative abundances in Tables S1 and S2 should be adjusted to represent percentages.

Response: Sorry for our mistake, we have corrected this mistake of Figure S1 according to your comments in our revised manuscript. I have rewritten the relative abundance in Table S1 and Table S2 to represent percentage. Please check these points in the revised manuscript.

4. There are instances where the manuscript is too verbose, repetitive, and confusing. For instance, lines 85-107 could be written more clearly to present the significance of metabolomics to the research. The reporting of the ACE and Chao indices are also confusing (lines 279-282).

Response: We have carefully reviewed our manuscript and have rewritten some sections to make the writing more concise and clearer. We have also corrected the reporting of the ACE and Chao

indexes in revised manuscript. Please check this point in the revision manuscript (Line 85-102 and Line 298-300).

5. On lines 182 and 184, two different sequencing platforms are noted.

Response: Thanks for pointing this out to us; we have made a correction to the manuscript. High-throughput sequencing analysis of bacterial and fungal rRNA genes was performed on the purified, pooled sample using the Illumina novaseq6000 at Biomarker Technologies Corporation (BMK), Beijing, China. There was one sequencing platform in this manuscript. Please check this point in the revised manuscript (Line 190-193).

Thanks again for your good suggestions and comments!

October 24, 2022

Dr. Xingxu Zhang
Lanzhou University
Tianshui nanlu, No.222
Lanzhou
China

Re: Spectrum01350-22R1 (The effect of *Epichloë gansuensis* endophyte on seed-borne microbes and seed metabolites in *Achnatherum inebrians*)

Dear Dr. Xingxu Zhang:

I and both reviewers think that the modifications made your manuscript better. The manuscript presents a substantial amount of data, but there are still an excessive number of correlations, and the manuscript needs further integration of your multiple results. Reducing the excessive number of correlations and making the manuscript text more straightforward by focusing on your main findings will help to make the manuscript clear and enable readers to get the real significance of your findings

Link Not Available

Sincerely,

Patricia Albuquerque

Journals Department
Reviewer comments:

Reviewer #1 (Comments for the Author):

I reviewed an earlier version of this manuscript where my major comment was that the authors analyzed a huge number of microbial and chemical variables, and calculated many hundreds or thousands of correlations among these variables that the issue of statistical tests and multiple comparisons arise. It was difficulty to understanding the major results given the sheer number pf comparisons and correlations.

In this revision, the authors try to address these issues but they still apply, although to a lesser extent. Also, the role of Epichloe as a member of the microbiota was not clearly distinguished from the other microbiota. Are most of the significant correlations due to Epichloe presence or absence. The study does represent a large investigation into of the dominant Epichloe symbiont affects the seed-borne microbiota and seed metabolites of drunken horse grass (DHG), and it uses state of the art sequencing and metabonomics technologies. The methodologies used to generate the data seem powerful and appropriate, and the authors have produced an enormous data set on the microbiota and metabolites.

The major issues I still have with this revision are the sheer number of correlations make it difficult to see the forest through the trees, the uncertainty about whether some or all of the metabolites are produced by the Epichloe fungus, the plant or other microbes, and the possible biological functions or the many metabolites.

Specific Comments:

Line 37 - unclear if these are fungal or plant origins? Line 55 too.

English grammar and composition needs improvement in several places (e.g., 37-40 and elsewhere).

38 - do the seed borne microbiota include Epichloe? How much variation does Epichloe account for vs. all of the other microbiota components?

43 - is there any take home message?

52 - not clear what LEp means. What is L representing?

56-57 - it sounds like an impressive number of correlations but the context could be clearer. How many correlations were examined in total? Also, if Epichloe is removed from the microbiota effects, how many significant correlations remain? This point is clarified to some extent later in the paper, but earlier it comes up as an obvious question.

105 - in relation to the above point, is Epichloe part of the microbiome or is it a driver of the microbiome (and the metabolites).

161 - would not some of the microbiota of glumes be the result of wind and spores blowing from one plot to the other so there should be a lot of overlap between E+ and E-?

240-241 - I don't understand what is happening here.

244 - it is unclear what the experimental factors for the ANOVA tests are? It seems that Epichloe status (+ or -) should be analyzed as a treatment effect. The description of statistics in the section needs to provide more information.

263 - see line 280-281. Can you say the same here?

294-303 - the differences between tissues and plant parts seems less interesting than the effect of Epichloe.

306 - remind us what these treatments are?

310 - clarify if these are plants metabolites, Epichloe metabolites or other microbial metabolites

324-326 - why the big difference between seeds and glumes?

335 - is methylergonovine an Epichloe product?

335-347 - This long list of compounds is less interesting without some knowledge of the biological functions of these compounds.

352-358 - are all of these enriched in E+ plants?

363 - does there need to be some more conservative statistical test given the large number of correlations. 5% of the correlations should be significant just by chance.

366 - 385 - this whole paragraph raises the issue whether the microbes produce the metabolites directly.

385 - 123 significant, how many correlations were tested?

366-451 - this long section of text makes it very difficult to understand what the major results are. There are so many combinations of microbes and metabolites, understanding what is biologically significant is difficult to see. I don't think that readers will go through this text in detail.

541 - is the conclusion that the alkaloids are fungal in origin? Then in 556-557, it is unclear what could be fungal origin and what could be plant origin. See 572-573 As well.

591-593 - there was little about functions mention in the extensive text on 366-451

Fig. 1 - some of the abbreviations do not have any intuitive sense.

Table 1 - this design needs to be clarified better in the text - that Epichloe is a treatment effect.

Reviewer #2 (Comments for the Author):

Summary:

The microbiome and metabolome of field-grown *Achnatherum inebrians* grass infected with the endophyte, *Epichloe gansuensis*, were studied. The significance of this research centers on the grass's invasiveness, which is correlated with the fungal endophyte. The conclusions that microbial diversity and metabolic profiles are correlated to endophyte infection are supported.

Major points:

Lines 386- 440 summarize the correlation data, but it reads as a list of data presented in Figure 4. These data represent the crux of their research study. Perhaps a network analysis would be more appropriate to show the integration of metabolites, microbial taxa, infection status, and type.

Minor points:

Minor editing is needed. For example,

Line 149 "...then stored in a -80 refrigerator" (missing °C)

Line 216 "...bacterial raw data is PRJNA865502. And the BioProject accession number of fungal.." These sentences need to be merged.

Lines 335-337 needs to be joined with sentence just prior to it.

Line 389 "...and Rokubacteria, Compared with SEF..." should have a period between the two sentences.

Staff Comments:

Preparing Revision Guidelines

Please return the manuscript within 60 days; if you cannot complete the modification within this time period, please contact me. If you do not wish to modify the manuscript and prefer to submit it to another journal, please notify me of your decision immediately so that the manuscript may be formally withdrawn from consideration by Microbiology Spectrum.

Reviewer #1 (Comments for the Author):

I reviewed an earlier version of this manuscript where my major comment was that the authors analyzed a huge number of microbial and chemical variables, and calculated many hundreds or thousands of correlations among these variables that the issue of statistical tests and multiple comparisons arise. It was difficult to understand the major results given the sheer number of comparisons and correlations.

In this revision, the authors try to address these issues but they still apply, although to a lesser extent. Also, the role of *Epichloë* as a member of the microbiota was not clearly distinguished from the other microbiota. Are most of the significant correlations due to *Epichloë* presence or absence. The study does represent a large investigation into of the dominant *Epichloë* symbiont affects the seed-borne microbiota and seed metabolites of drunken horse grass (DHG), and it uses state of the art sequencing and metabolomics technologies. The methodologies used to generate the data seem powerful and appropriate, and the authors have produced an enormous data set on the microbiota and metabolites.

The major issues I still have with this revision are the sheer number of correlations make it difficult to see the forest through the trees, the uncertainty about whether some or all of the metabolites are produced by the *Epichloë* fungus, the plant or other microbes, and the possible biological functions of the many metabolites.

We sincerely thank you for the very thorough and informed revision of our manuscript and the important concerns and comments that you have made. We have carefully studied what you have indicated and have endeavored to make the improvements required.

We again analyzed correlations between seed differential metabolites and seed-borne microbial phyla via network analysis. This revision has simpler content, a clearer Figure 4, and more comprehensible results than the original manuscript. Please check these changes in Figure 4, Table S8-S13, and Line 392-437 of our revised manuscript.

In our manuscript, we detected metabolites produced by EI/EF seeds and glumes, and found that compared with EF plants, the presence of *Epichloë gansuensis* resulted in changes in the content of 108 seed and 31 glume differential metabolites in EI plants. We focused on differences of metabolites between EI and EF plants, and didn't detect metabolites produced by *Epichloë* endophyte and other microbes. We think that the metabolites detected in this study were produced

by the plant, both when the *Epichloë* endophyte was present, namely a plant response, or when absent.

Purposes of this manuscript were to confirm the effects of *Epichloë* endophyte on communities and diversities of seed-borne microbes, and to study the content and classes of seed metabolites. Biological functions of seed-borne microbes and seed metabolites were not studied. However, we performed correlation network analysis for analyzing relationships between seed differential metabolites and microbial phyla, and found there were many important relationships between highly connected microbes like Actinobacteria, Acidobacteria and Ascomycota, and highly connected metabolites such as Tyr-Met, myristic acid and 6-Hydroxymelatonin. Further research would focus on biological function of these metabolites or microbes.

Specific Comments:

Line 37 - unclear if these are fungal or plant origins? Line 55 too.

Response: Sorry, we didn't make it clear. We rewrote these sentences for these metabolites to make the information clear that these were plant products. Please check these changes in our revised manuscript for Line 31-33 and Line 51-52.

English grammar and composition needs improvement in several places (e.g., 37-40 and elsewhere).

Response: We checked the manuscript carefully and corrected many mistakes.

38 - do the seed borne microbiota include *Epichloë*? How much variation does *Epichloë* account for vs. all of the other microbiota components?

Response: Yes, the seed-borne microbiota of endophyte-infected plants does include the *Epichloë* species. We added analysis for the relative abundance of other fungi of the eight samples at the genus level in the presence or absence of *Epichloë*, for detailed information please see Figure 1. The average of relative abundance of *Epichloë* endophyte and other fungi in the eight SEInI samples was 10.95% and 89.05%, respectively. Meanwhile, the relative abundance of *Epichloë* endophyte in 8 SEPI samples were 0.14%, 1.28%, 0.10%, 0.09%, 0.00%, 0.04%, 1.97% and 0.43%, and relative abundance of other fungi were 99.86%, 98.72%, 99.90%, 99.91%, 100.00%, 99.96%, 98.03% and 99.57%, the average of relative abundance of *Epichloë* endophyte and other fungi in 8 seed epiphytic fungi of endophyte-infected plants were 0.51% and 99.49%, respectively.

Figure 1. The relative abundance of seed-borne microbial communities at the genus level.

Relative abundance of *Epichloë* and other fungi of all samples at the genus level (SepF: seed epiphytic fungi of endophyte-free plants, SEpI: seed epiphytic fungi of endophyte-infected plants, SEnF: seed endophytic fungi of endophyte-free plants, SEnI: seed endophytic fungi of endophyte-infected plants, GEpF: glume epiphytic fungi of endophyte-free plants, GEpI: glume epiphytic fungi of endophyte-infected plants, n=8).

43 - is there any take home message?

Response: We added a take-home sentence that represents concentrated results of this study at the end of the abstract and the discussion.

52 - not clear what LEp means. What is L representing?

Response: Sorry, we didn't make it clear. LEp represented the glumes epiphytic microbe, the LEp was changed to "GEp", "G" represented glume and "Ep" represented epiphytic. We have corrected this mistake and added explanation of GEp according to your comments in our revised manuscript. Please see it in Line 49.

56-57 - it sounds like an impressive number of correlations but the context could be clearer. This point is clarified to some extent later in the paper, but earlier it comes up as an obvious question.

Response: We redid analyze for correlations between seed differential metabolites and seed-borne microbial phyla via network analysis, this revision had simpler content and clearer Figure 4, please check these changes in Figure 4, Table S8-S13, and Line 392-437 in results of our revised

manuscript.

105 - in relation to the above point, is *Epichloë* part of the microbiome or is it a driver of the microbiome (and the metabolites).

Response: *Epichloë* is both part of the microbiome and a driver of the microbiome and metabolites, the presence of *Epichloë* endophyte altered communities and diversities of seed-borne microbes, and altered content and classes of seed metabolites in this study.

161 - would not some of the microbiota of glumes be the result of wind and spores blowing from one plot to the other so there should be a lot of overlap between E+ and E-?

Response: The microbiota of glumes were the result of wind and spores blowing from one plot to the other, this viewpoint also was proved by Nelson, (2018). For analyzing seed-borne microbes, less research focused on glume microbes, our manuscript was an important supplement for glumes microbes and seed-borne microbes. Microbes overlapped between EI (endophyte-infected) and EF (endophyte-free) represented core seed-borne microbiome of *A. inebrians*, this result was illustrated by venn diagram in Figure S2 of our revised manuscript (see it as following).

Reference:

1. Nelson EB. The seed microbiome: Origins, interactions, and impacts. Plant Soil. 2018, 422, 7-34.

Figure S2. Venn diagram within seed and glume bacterial (A) and fungal (B) OTUs of EI/EF seeds and glumes of *A. inebrians* (n=8).

240-241 - I don't understand what is happening here.

Response: This question was in Line 250 of our revised manuscript. Sorry for my mistake, I have corrected it in our revised manuscript. The project (VIP, value of OPLS-DA model)>1, P

value<0.05, and fold change (FC)>2 is the selection criteria of seed and glume differential metabolites in this manuscript, based on the FC>2, VIP>1 and P<0.05, 108 and 31 differential metabolites between EI vs EF seeds and glumes were separated, respectively (Table S6-S7 of revised manuscript), Fu et al. (2020), Zhang et al. (2021) and Huang et al. (2022) also used this selection criteria to analyze differential metabolites.

Reference:

1. Fu M, Zhang XW, Liang YH, Lin SR, Qian WP, Fan SR. Alterations in vaginal microbiota and associated metabolome in women with recurrent implantation failure. *mBio*. 2020, 11, e03242-19.
2. Zhang J, Feng D, Law HKW, Wu Y, Zhu GH, Huang WY, Kang Y. Integrative analysis of gut microbiota and fecal metabolites in rats after prednisone treatment. *Microbiology Spectrum*. 2021, 9, e00650-21.
3. Huang K, Wang YG, Bai Y, Luo QY, Lin XC, Yang QY, Wang SH, Xin HJ. Gut microbiota and metabolites in atrial fibrillation patients and their changes after catheter ablation. *Microbiology Spectrum*. 2022, 10, e01077-21.

244 - it is unclear what the experimental factors for the ANOVA tests are? It seems that *Epichloë* status (+ or -) should be analyzed as a treatment effect. The description of statistics in the section needs to provide more information.

Response: We supplemented detailed experimental factors for the ANOVA tests involved in this manuscript, check it in statistical analyses part. Please see it in Line 269-277.

The infection status (EF: endophyte-free, EI: endophyte-infected) of the *E. gansuensis* endophyte was one of the treatments involved in our manuscript; another treatment was different parts of EI and EF seeds. Detailed grouping information of treatments showed in “Treatments information of seed-borne microbes and seed metabolites” of material and methods, please check it in Line 253 of revised manuscript.

Some detailed information was added to the description of statistical analyses, please check it in Line 267-282 of revised manuscript.

263 - see line 280-281. Can you say the same here?

Response: Sorry, we didn't make it clear. We corrected these points to make it clear. Please check it in Line 298 and Line 316-317.

294-303 - the differences between tissues and plant parts seems less interesting than the effect of

Epichloë.

Response: Sorry, we didn't make it clear. We simplified the description for the effect of different parts of EI and EF seeds on seed-borne microbial diversities; meanwhile, we supplied the description for the effect of the infection status of the *Epichloë gansuensis* endophyte on seed-borne microbial diversities. Please see it in Line 330-335.

306 - remind us what these treatments are?

Response: Thanks for pointing out this problem, please check its changes in Line 338.

There were two treatments involved in our manuscript, one was the infection status of the *E. gansuensis* endophyte, and another was different parts of EI and EF seeds. Detailed grouping information of treatments showed in "Treatments information of seed-borne microbes and seed metabolites" of material and methods, please see it in Line 253.

310 - clarify if these are plants metabolites, *Epichloë* metabolites or other microbial metabolites

Response: In our manuscript, we analyzed the metabolites produced by seeds and glumes of EI and EF plants, and found that compared with EF plants, EI plants had many differential metabolites of seed and glume. In the present study, we focused on the difference of metabolites between EI and EF plants, and we didn't detect metabolites produced by *E. gansuensis* endophyte or other microbes in EF plants. So we think these metabolites involved in our manuscript were produced by the interaction between endophyte and plants.

324-326 - why the big difference between seeds and glumes?

Response: This question was in Line 356-358 of our revised manuscript. The glumes are modified leaves that enclose the seed and are so very different in exposure to air-borne fungi and bacteria than the protected seeds. Further, the glumes become very dry and probably cease being metabolically active, whereas the seeds will retain metabolic activity, especially within the embryo, scutellum and endosperm.

335 - is methylergonovine an *Epichloë* product?

Response: Methylergonovine wasn't an *Epichloë* product, as this product was found in both EI and EF seeds.

335-347 - This long list of compounds is less interesting without some knowledge of the biological functions of these compounds.

Response: The aims of this manuscript were to analyze the effects of *Epichloë* endophyte on

communities and diversities of seed-borne microbes, and on content and classes of seed metabolites - we didn't analyze biological function of seed-borne microbes and seed metabolites. However, we performed a correlation network analysis for differential metabolites and microbial phyla levels, and further research will focus on biological function of seed-borne microbes and seed metabolites.

352-358 - are all of these enriched in E+ plants?

Response: These results were updated in our revised manuscript for Line 385-391, and all of these metabolites pathways enriched in EI (*Epichloë* endophyte-infected) plants, these results are shown in Table S4-S5.

363 - does there need to be some more conservative statistical test given the large number of correlations. 5% of the correlations should be significant just by chance.

Response: We analyzed again correlations between seed differential metabolites and seed-borne microbial phyla via network analysis, please check these results in Figure 4, Table S8-S13, and Line 392-437 of our revised manuscript. About 5% of the correlations were referred to Noecker et al. (2019), Fu et al. (2020) and Huang et al. (2022).

Reference:

1. Noecker C, Chiu HC, McNally CP, et al. Defining and evaluating microbial contributions to metabolite variation in microbiome-metabolome association studies. *mSystems*. 2019, 4, e00579-19.
2. Fu M, Zhang XW, Liang YH, et al. Alterations in vaginal microbiota and associated metabolome in women with recurrent implantation failure. *mBio*. 2020, 11, e03242-19.
3. Huang K, Wang YG, Bai Y, et al. Gut microbiota and metabolites in atrial fibrillation patients and their changes after catheter ablation. *Microbiol Spectr*. 2022, 10, e01077-21.

366 - 385 - this whole paragraph raises the issue whether the microbes produce the metabolites directly.

Response: This question was in Line 397-414 of our revised manuscript. In our manuscript, we analyzed the effects of *Epichloë* endophyte on communities and diversities of seed-borne microbes, and on content and classes of seed metabolites, found there were many close and complex relationships between metabolites and microbes. We focused on the difference of metabolites between EI and EF plants, and didn't detect metabolites produced by *Epichloë*

endophyte or other microbes. We thus think that metabolites detected in this study were produced by the interaction between endophyte and plants.

385 - 123 significant, how many correlations were tested?

Response: In the original manuscript, total of 3160 correlations between 316 seed differential metabolites and 10 seed endophytic bacterial phyla, among them, there were 123 significant ($P < 0.05$) correlations. This result was updated in our revised manuscript via using network analysis, please check it in Line 397.

366-451 - this long section of text makes it very difficult to understand what the major results are. There are so many combinations of microbes and metabolites, understanding what is biologically significant is difficult to see. I don't not think that readers will go through this text in detail.

Response: We redid analyze about correlations network between seed differential metabolites and seed-borne microbial phyla via network analysis, meanwhile, we simplified content of results and cleared the Figure 4 - please check these results in Figure 4, Table S8-S13, and Line 392-437 of our revised manuscript.

In this manuscript, we analyzed the effects of *E. gansuensis* endophyte on communities and diversities of seed-borne microbes, and on content and classes of seed metabolites. We didn't analyze biological function of these microbes and metabolites, however, we calculated the relationships between seed-borne microbial phyla and seed differential metabolites, and found many important relationships between high connected microbes like Actinobacteria, Acidobacteria and Ascomycota, and high connected metabolites such as Tyr-Met, myristic acid and 6-Hydroxymelatonin. Further research will focus on biological function of these metabolites or microbes.

541 - is the conclusion that the alkaloids are fungal in origin? Then in 556-557, it is unclear what could be fungal origin and what could be plant origin. See 572-573 As well.

Response: Gao (2006) found that alkaloids were produced by *Epichloë* endophyte culturing in medium, the content of alkaloids detected in the leaves of *A. inebrians* plants infected by *Epichloë* endophyte was 35 or 37 times higher than produced by *Epichloë* endophytic fungi under in medium culture conditions (Li, 2005; Gao, 2006).

In our manuscript, we analyzed the effects of *Epichloë* endophyte on communities and diversities of seed-borne microbes, content and classes of seed metabolites, found there were many close and

complex relationships between microbes and metabolites. We focused on the difference between EI and EF plants, didn't detected metabolites produced by *Epichloë* endophyte and other microbes, so we think that metabolites involved in this manuscript were produced by interactions between endophyte and plants. We calculated correlations between seed-borne microbial phyla and seed differential metabolites via network analysis, further researches would focused on analyzing metabolites produced by *Epichloë* endophyte or other microbes.

Reference:

1. Gao JH. Detection of ergot alkaloid production by *Neotyphodium gansuensis* in two and sequencing the *dmaW* gene of *N. lolii* isolates AR1 and Lp19. Lanzhou University, 2006, Lanzhou. (In Chinese with English abstract).
2. Li CJ. Biological and Ecological Characteristics of *Achnatherum inebrians/Neotyphodium* endophyte symbiont. Lanzhou University, Lanzhou. (In Chinese with English abstract).

591-593 - there was little about functions mention in the extensive text on 366-451

Response: Our manuscript analyzed the effects of *E. gansuensis* endophyte on communities and diversity of seed-borne microbes and content and classes of seed metabolites. We focused on the difference of metabolites involved in this manuscript, and didn't analyze function of these metabolites. We calculated the relationships between seed microbes and seed metabolites via network analysis; further research will focus on biological function of these metabolites and microbes.

Fig. 1 - some of the abbreviations do not have any intuitive sense.

Response: Sorry, we didn't make it clear. We revised the expression for these abbreviations of grouping treatments in material and methods part for "Treatments information of seed-borne microbes and seed metabolites", please see it in Line 253. And these abbreviations were explained carefully in figures and tables, please check these changes in our revised manuscript.

Table 1 - this design needs to be clarified better in the text - that *Epichloë* is a treatment effect.

Response: We supplied grouping of two treatments (the infection status of the *Epichloë gansuensis* endophyte and different parts of EI and EF seeds) in material and methods, please see the section "Treatments information of seed-borne microbes and seed metabolites" in our revised manuscript. Please check it in Line 253.

Reviewer #2 (Comments for the Author):

Summary:

The microbiome and metabolome of field-grown *Achnatherum inebrians* grass infected with the endophyte, *Epichloë gamsuensis*, were studied. The significance of this research centers on the grass's invasiveness, which is correlated with the fungal endophyte. The conclusions that microbial diversity and metabolic profiles are correlated to endophyte infection are supported.

Major points:

Lines 386- 440 summarize the correlation data, but it reads as a list of data presented in Figure 4. These data represent the crux of their research study. Perhaps a network analysis would be more appropriate to show the integration of metabolites, microbial taxa, infection status, and type.

Response: We analyzed again correlations between seed differential metabolites and seed-borne microbial phyla via network analysis, please check these results in Figure 4, Table S8-S13, and Line 392-437 of our revised manuscript.

Minor points:

Minor editing is needed. For example,

Line 149 "...then stored in a -80 refrigerator" (missing °C)

Response: We added "°C" after -80. Please see in Line 147 in our revised manuscript.

Line 216 "...bacterial raw data is PRJNA865502. And the BioProject accession number of fungal."

These sentences need to be merged.

Response: These sentences were merged, Please see in Line 210 in our revised manuscript.

Lines 335-337 needs to be joined with sentence just prior to it.

Response: This sentence was joined with the prior sentence. Please see in Line 347 in our revised manuscript.

Line 389 "...and Rokubacteria, Compared with SEF..." should have a period between the two sentences.

Response: Sorry for my mistake, I corrected it. Please check it in Line 401 in our revised manuscript.

Thanks again for your good suggestions and comments!

December 17, 2022

Dr. Xingxu Zhang
Lanzhou University
Tianshui nanlu, No.222
Lanzhou
China

Re: Spectrum01350-22R2 (The effect of *Epichloë gansuensis* endophyte on seed-borne microbes and seed metabolites in *Achnatherum inebrians*)

Dear Dr. Xingxu Zhang:

While we are willing to consider a revised version of this paper at Spectrum, it would be in your best interest to improve the writing. I recommend that you ask a colleague of yours who is a native English speaker to read and provide you some feedback on the writing. You are also welcome to use one of the services here: <https://journals.asm.org/content/language-editing-services>

Thank you for submitting your manuscript to Microbiology Spectrum. As you will see your paper is very close to acceptance. Please modify the manuscript along the lines I have recommended. As these revisions are quite minor, I expect that you should be able to turn in the revised paper in less than 30 days, if not sooner. If your manuscript was reviewed, you will find the reviewers' comments below.

When submitting the revised version of your paper, please provide (1) point-by-point responses to the issues raised by the reviewers as file type "Response to Reviewers," not in your cover letter, and (2) a PDF file that indicates the changes from the original submission (by highlighting or underlining the changes) as file type "Marked Up Manuscript - For Review Only". Please use this link to submit your revised manuscript. Detailed instructions on submitting your revised paper are below.

Link Not Available

Sincerely,

Patricia Albuquerque

Reviewer comments:

Reviewer #2 (Comments for the Author):

The authors are attempting to describe complex correlations between metabolomes and microbiomes (fungal and bacterial) of three treatment groups (endophyte-infected or not, and epiphytic), and two anatomical structures of the grass (glumes and seeds). With the amount of data presented, it can sometimes be difficult to recall the significance of this project. Having reviewed an earlier version of this manuscript, I appreciate the network analysis and clarifications that were included in this current version. Significant editing is still needed, as noted below.

Lines 147 and 169: -80C freezer (not refrigerator)

Line 149: The use of "And" to begin a sentence should be corrected. (Noted in this line and others throughout the manuscript)

Line 186: NovaSeq is capitalized.

Line 198: Consider changing this line to read "The alpha diversity indices that were calculated included..".

Line 210: Consider changing this line to read "...raw data are...".

Line 212: "...six samples of EI..."

Line 213: "...for the detection of seed..."

Line 216: "...1) containing an isotopically-labelled..."

Line 217: "After 30 s of vortexed..."

Line 253: Consider changing the title of this section. I appreciate this was included as it helped clarify the various treatments of the plants.

Line 268: Include "and" between SigmaPlot and R software.

Line 296: It is unclear what you mean by reaches the sequencing depth. Do you mean maximum sequencing depth?

In the section beginning on line 313, it can be simplified by removing statistical values. They are not included for all comparisons anyway.

Line 331: "...effects on seed-fungal diversities...."

Lines 368 and 394, Table 3 and 4 titles: Rather than up and down accumulated, consider using differentially accumulated.

Table 1: Please include a dividing line in the table between the Bacteria and Fungi values.

The description of what "E" and "P" mean (in Table 2 legend) would be really helpful if also included in Figure 2 legend.

Line 393: Consider changing microorganism to microbial taxa.

Line 394: Consider changing to "...correlation networks between metabolites that were significantly up- or down-regulated in seeds..." Perhaps eliminate "...and found that....relationships between them."

Lines 403-422: Please clarify what you mean by "degree". For instance, in line 405 do you mean that there were 48 and 35 positive correlations between Acidobacteria and Actinobacteria with the metabolites? This section also has several grammatical issues that need to be corrected.

Line 410: What is meant by whole bacterial and fungal networks? Do you mean both networks combined?

Line 415: "....between differentially accumulated metabolites...."

Reviewer #3 (Comments for the Author):

The present study investigated the influence of *E. gansuensis* endophyte on seed-borne microbial communities and seed metabolites of DHG. They performed the correlation analysis between seed-borne microbes and seed metabolites and found many close and complex correlations between them. The results and statistical methods were confidential.

1.Highlight should be a statement of the main conclusions, not a list of general statistics. epiphytic fungi of glumes (GEp) was never to appear again in Highlight, so the " (GEp)" should be deleted. While "EF" and "EI" should be written in full name.

2.Line 28, the abbreviation of "*E. gansuensis*" appears at the first time, it needs to be marked in parentheses. The same goes for LC/MS.

3.Line148-150, line 212, the format of the quantifier needs to be consistent. Please check the full text carefully.

4.Line 188-196, ASV has more obvious advantages over OTU, and there are many articles in the form of ASV. If ASV is adopted in this paper, the analysis results may be more reliable and get better analysis and conclusions.

5.Line 300, (493 OTUs 58% sequences) should be (493 OTUs, 58% sequences)

6.Figure S1-S6, pictures should be clear, with the same font size and type, and no Chinese characters. Please check and revise carefully.

7.One stylistic issue was that many statements were written in the active voice, using "we" or "Our" as the subject pronoun.

Although sentences beginning with a subject pronoun have their place, many of the sentences in a scientific manuscript are more effectively stated in a passive voice, beginning with a subject noun (for emphasis), not a pronoun. Besides, the language of this article needs further polishing.

Overall this a good research paper and if presented properly will make an interesting contribution to this subject.

Preparing Revision Guidelines

Please return the manuscript within 60 days; if you cannot complete the modification within this time period, please contact me. If you do not wish to modify the manuscript and prefer to submit it to another journal, please notify me of your decision immediately so that the manuscript may be formally withdrawn from consideration by Microbiology Spectrum.

Summary of comments

The present study investigated the influence of *E. gansuensis* endophyte on seed-borne microbial communities and seed metabolites of DHG. They performed the correlation analysis between seed-borne microbes and seed metabolites and found many close and complex correlations between them. The results and statistical methods were confidential.

1. Highlight should be a statement of the main conclusions, not a list of general statistics. epiphytic fungi of glumes (GEp) was never to appear again in Highlight, so the “ (GEp)” should be deleted. While “EF” and “EI” should be written in full name.
2. Line 28, the abbreviation of “*E. gansuensis*” appears at the first time, it needs to be marked in parentheses. The same goes for LC/MS.
3. Line148-150, line 212, the format of the quantifier needs to be consistent. Please check the full text carefully.
4. Line 188-196, ASV has more obvious advantages over OTU, and there are many articles in the form of ASV. If ASV is adopted in this paper, the analysis results may be more reliable and get better analysis and conclusions.
5. Line 300, (493 OTUs 58% sequences) should be (493 OTUs, 58% sequences)
6. Figure S1-S6, pictures should be clear, with the same font size and type, and no Chinese characters. Please check and revise carefully.
7. One stylistic issue was that many statements were written in the active voice, using “we” or “Our” as the subject pronoun. Although sentences beginning with a subject pronoun have their place, many of the sentences in a scientific manuscript are more effectively stated in a passive voice, beginning with a subject noun (for emphasis), not a pronoun. Besides, the language of this article needs further polishing.

Overall this a good research paper and if presented properly will make an interesting contribution to this subject.

Reviewer #2 (Comments for the Author):

The authors are attempting to describe complex correlations between metabolomes and microbiomes (fungal and bacterial) of three treatment groups (endophyte-infected or not, and epiphytic), and two anatomical structures of the grass (glumes and seeds). With the amount of data presented, it can sometimes be difficult to recall the significance of this project. Having reviewed an earlier version of this manuscript, I appreciate the network analysis and clarifications that were included in this current version. Significant editing is still needed, as noted below.

We sincerely thank you for the very thorough and informed revision of our manuscript and the important concerns and comments that you have made. We have carefully studied what you have indicated and have endeavored to make the improvements required.

Lines 147 and 169: -80C freezer (not refrigerator)

Response: Sorry for my mistake, I corrected it. Please check it in Line 148 and 170 in our revised manuscript.

Line 149: The use of "And" to begin a sentence should be corrected. (Noted in this line and others throughout the manuscript)

Response: Sorry for my mistake, I corrected this question throughout the manuscript. Please check it in the revised manuscript.

Line 186: NovaSeq is capitalized.

Response: Sorry for my mistake, I corrected it. Please see it in Line 186.

Line 198: Consider changing this line to read "The alpha diversity indices that were calculated included..".

Response: Thanks for your good suggestion, we corrected this sentence, please check it in Line 198.

Line 210: Consider changing this line to read "...raw data are...".

Response: We corrected it, please check it in Line 210.

Line 212: "...six samples of EI..."

Response: We corrected it, please check it in Line 212.

Line 213: "...for the detection of seed..."

Response: We corrected this sentence, please check it in Line 213.

Line 216: "...1) containing an isotopically-labelled..."

Response: We corrected this sentence, please check it in Line 217.

Line 217: "After 30 s of vortexed..."

Response: We corrected this sentence, please check it in Line 218.

Line 253: Consider changing the title of this section. I appreciate this was included as it helped clarify the various treatments of the plants.

Response: We changed the title of this section, please check it in Line 255.

Line 268: Include "and" between SigmaPlot and R software.

Response: We corrected this sentence, please check it in Line 270.

Line 296: It is unclear what you mean by reaches the sequencing depth. Do you mean maximum sequencing depth?

Response: Yes, from Figure S3, we can see that the OTU number was not bigger with the number of changing sequences sampled and the sequencing depths represented the maximum sequencing depth. When the curve tends to be flat, it indicated that the amount of sequencing data is reasonable, and more data will only generate a few new species; otherwise, it indicates that more new species may be generated by continued sequencing, which can reflect the majority of microbial diversity information in the samples.

Figure S3. Rarefaction curves of the 16S rRNA and ITS gene sequence from the seed and glume bacteria (A) and fungi (B) in and on seeds and glumes of EI and EF *Achnatherum inebrians*, (n=8).

In the section beginning on line 313, it can be simplified by removing statistical values. They are not included for all comparisons anyway.

Response: We removed statistical values of this section “Seeds and glumes bacterial and fungal community diversity”, please check it in Line 316-339.

Line 331: "...effects on seed-fungal diversities...."

Response: We corrected this sentence, please check it in Line 330.

Lines 368 and 394, Table 3 and 4 titles: Rather than up and down accumulated, consider using differentially accumulated.

Response: We corrected these points, please check these in Line 366, Table 3 and 4 titles.

Table 1: Please include a dividing line in the table between the Bacteria and Fungi values.

Response: We added a dividing line in the table between the bacteria and fungi values. Please see it in Table 1.

The description of what "E" and "P" mean (in Table 2 legend) would be really helpful if also included in Figure 2 legend.

Response: We added the description of what "E" and "P" mean in Figure 2 legend.

Line 393: Consider changing microorganism to microbial taxa.

Response: We corrected it, please check it in Line 393.

Line 394: Consider changing to "...correlation networks between metabolites that were significantly up- or down-regulated in seeds..." Perhaps eliminate "...and found that....relationships between them."

Response: We corrected this sentence as your good suggestion, and eliminate "...and found that....relationships between them." Please check it in Line 392-393.

Lines 403-422: Please clarify what you mean by "degree". For instance, in line 405 do you mean that there were 48 and 35 positive correlations between Acidobacteria and Actinobacteria with the metabolites? This section also has several grammatical issues that need to be corrected.

Response: We added the mean of degree in Line 402-403, and checked the manuscript carefully and corrected many grammatical mistakes.

Line 410: What is meant by whole bacterial and fungal networks? Do you mean both networks combined?

Response: Yes, in Line 410, whole bacterial and fungal networks means all correlations were analyzed both in bacterial and fungal networks.

Line 415: "...between differentially accumulated metabolites...."

Response: We corrected this point, please check it in Line 413-414.

Reviewer #3 (Comments for the Author):

The present study investigated the influence of *E. gansuensis* endophyte on seed-borne microbial communities and seed metabolites of DHG. They performed the correlation analysis between seed-borne microbes and seed metabolites and found many close and complex correlations between them. The results and statistical methods were confidential.

1. Highlight should be a statement of the main conclusions, not a list of general statistics. epiphytic fungi of glumes (GEp) was never to appear again in Highlight, so the " (GEp)" should be deleted. While "EF" and "EI" should be written in full name.

Response: Thanks for your good suggestion. We have deleted the “GEP”, and have changed the EF and EI to endophyte-free and endophyte-infected, respectively. Please see it in Line 50, 52-53.

2. Line 28, the abbreviation of "*E. gansuensis*" appears at the first time, it needs to be marked in parentheses. The same goes for LC/MS.

Response: In our understanding, once a species name is written containing both the generic and species name from then on the generic name can be abbreviated to the capital letter of the name. As the full name *Epichloë gansuensis* was given in Line 25 of the first sentence from then on we have written the genus name of species of this genus as E. species-name. We feel that as LC-MS is such a well-defined technology and “LC-MS” is in brackets after the full name when first written in the abstract (Line 31), highlights (Line 47), and the Introduction (Line 97).

3. Line 148-150, line 212, the format of the quantifier needs to be consistent. Please check the full text carefully.

Response: We checked and corrected the format of the quantifier, please check it in Line 212-215.

4. Line 188-196, ASV has more obvious advantages over OTU, and there are many articles in the form of ASV. If ASV is adopted in this paper, the analysis results may be more reliable and get better analysis and conclusions.

Response: ASV has more obvious advantage over OTU, but ASV is an important prerequisite that our data can establish a suitable error model and accurately detect the wrong sequence. Our data was detected in 2019, the sequencing company and platform provided OTUs for our sequencing data rather than ASV because of the poor error model in 2019. Recently, ASV was extensive used since 2021, like Bernard et al. (2021), Ling et al. (2022) and other studies. But we still can find that OTU also used in many studies, like Huang et al., 2019, Wei et al. (2020), Yuan et al. (2021),

Wu et al. (2022) and Liu et al. (2022).

References:

1. Bernard J, Wall CB, Costantini MS, et al. Plant part and a steep environmental gradient predict plant microbial composition in a tropical watershed[J]. ISME Journal. 2021, 15, 999-1009.
2. Ling N, Wang TT, Kuzyakov Y. Rhizosphere bacteriome structure and functions[J]. Nature Communication, 2022, 13, 836.
3. Huang FJ, Zheng XJ, Ma XH, et al. Theabrownin from Pu-erh tea attenuates hypercholesterolemia via modulation of gut microbiota and bile acid metabolism[J]. Nature Communication. 2019, 10, 4971.
4. Wen T, Zhao ML, Liu T, et al. High abundance of *Ralstonia solanacearum* changed tomato rhizosphere microbiome and metabolome[J]. BMC Plant Biology. 2020, 20, 166.
5. Yuan MM, Guo X, Wu LW, et al. Climate warming enhances microbial network complexity and stability[J]. Nature Climate Change. 2021, 11, 343-348.
6. Wu MH, Xue K, Wei PJ, et al. Soil microbial distribution and assembly are related to vegetation biomass in the alpine permafrost regions of the Qinghai-Tibet Plateau[J]. Science of the Total Environment. 2022, 834, 155259.
7. Liu BW, Ju YW, Xia C, et al. The effect of *Epichloë* endophyte on phyllosphere microbes and leaf metabolites in *Achnatherum inebrians*[J]. iScience. 2022, 25, 104144.

5.Line 300, (493 OTUs 58% sequences) should be (493 OTUs, 58% sequences)

Response: We have corrected it, please check in Line 302.

6.Figure S1-S6, pictures should be clear, with the same font size and type, and no Chinese characters. Please check and revise carefully.

Response: We carefully checked and corrected these pictures, please check it.

7.One stylistic issue was that many statements were written in the active voice, using "we" or "Our" as the subject pronoun. Although sentences beginning with a subject pronoun have their place, many of the sentences in a scientific manuscript are more effectively stated in a passive voice, beginning with a subject noun (for emphasis), not a pronoun. Besides, the language of this article needs further polishing.

Response: We carefully checked and corrected these issues throughout the present study, please check it in our revised manuscript. We also polished the language of this article, thanks for your

suggestion.

Thanks again for your good suggestions and comments!

January 23, 2023

Dr. Xingxu Zhang
Lanzhou University
Tianshui nanlu, No.222
Lanzhou
China

Re: Spectrum01350-22R3 (The effect of *Epichloë gansuensis* endophyte on seed-borne microbes and seed metabolites in *Achnatherum inebrians*)

Dear Dr. Xingxu Zhang:

Your manuscript has been accepted, and I am forwarding it to the ASM Journals Department for publication. You will be notified when your proofs are ready to be viewed.

Sincerely,

Patricia Albuquerque
Editor, Microbiology Spectrum
